# VLMLight: Safety-Critical Traffic Signal Control via Vision-Language Meta-Control and Dual-Branch Reasoning Architecture

**Maonan Wang**[1,2*]   **Yirong Chen**[2*]   **Aoyu Pang**[1]   **Yuxin Cai**[3]
**Chung Shue Chen**[4]   **Yuheng Kan**[5]   **Man-On Pun**[1†]

[1]The Chinese University of Hong Kong, Shenzhen, China
[2]Shanghai AI Laboratory, Shanghai, China
[3]Nanyang Technological University, Singapore
[4]Nokia Bell Labs, Paris, France
[5]Fourier Intelligence, Shanghai, China

{maonanwang,aoyupang}@link.cuhk.edu.cn   chenyirong@pjlab.org.cn
caiy0039@e.ntu.edu.sg   chung_shue.chen@nokia-bell-labs.com
kanyuheng@gmail.com   simonpun@cuhk.edu.cn

## Abstract

Traffic signal control (TSC) is a core challenge in urban mobility, where real-time decisions must balance efficiency and safety. Existing methods—ranging from rule-based heuristics to reinforcement learning (RL)—often struggle to generalize to complex, dynamic, and safety-critical scenarios. We introduce **VLMLight**, a novel TSC framework that integrates vision-language meta-control with dual-branch reasoning. At the core of VLMLight is the first image-based traffic simulator that enables multi-view visual perception at intersections, allowing policies to reason over rich cues such as vehicle type, motion, and spatial density. A large language model (LLM) serves as a safety-prioritized meta-controller, selecting between a fast RL policy for routine traffic and a structured reasoning branch for critical cases. In the latter, multiple LLM agents collaborate to assess traffic phases, prioritize emergency vehicles, and verify rule compliance. Experiments show that VLMLight reduces waiting times for emergency vehicles by up to **65%** over RL-only systems, while preserving real-time performance in standard conditions with less than **1%** degradation. VLMLight offers a scalable, interpretable, and safety-aware solution for next-generation traffic signal control.

## 1   Introduction

Efficient management of urban traffic is a critical global challenge, with congestion leading to significant economic losses, environmental damage, and a decreased quality of life [1]. Traffic signal control (TSC) at intersections plays an essential role in regulating traffic flow and reducing delays [2]. Traditional TSC methods, such as Webster's method [3], MaxPressure control [4], and Self-Organizing Traffic Lights (SOTL) [5], rely on predefined rules and domain-specific heuristics. While reliable under steady conditions, these rule-based approaches cannot cope with the dynamic, stochastic, and non-stationary realities of modern road networks. As a result, they struggle to respond to unexpected changes in traffic patterns, where real-time, safety-critical decision-making is required to prevent delays and ensure public safety.

---

*Equal contribution
†Corresponding author: simonpun@cuhk.edu.cn

39th Conference on Neural Information Processing Systems (NeurIPS 2025).

To overcome the rigidity of rule-based approaches, recent studies have explored reinforcement learning (RL) as a more flexible paradigm for TSC [6, 7, 8, 9, 10]. By interacting with the environment and learning from feedback, RL agents can dynamically adjust signal phases in response to changing traffic conditions, offering improved adaptability and long-term optimization. However, despite their success in routine scenarios, these methods still rely on simplified, vectorized state representations, such as queue lengths [11, 12, 13] or intersection pressure [14, 15], and use static reward formulations [16, 17]. Such abstractions omit the semantically rich cues needed for context-dependent decisions. In safety-critical situations, such as granting the right-of-way to ambulances, RL agents trained solely on minimizing delay or maximizing throughput can deviate from the real-world priorities. Although some research [18, 19] has started to consider special vehicles, these methods still face two key challenges: first, determining the trade-off between emergency and regular traffic; second, integrating these priority mechanisms with existing RL-based controllers that differ in agent architecture and reward formulation. Consequently, current RL approaches still fall short when rapid, semantically informed intervention is required.

The emergence of large language models (LLMs) [20] has opened new possibilities for incorporating high-level reasoning and generalization into traffic signal control systems [21, 22]. Recent efforts have explored the use of LLMs to handle edge cases and long-tail scenarios that are difficult for RL-based policies to address [23, 24, 25]. However, current LLM-based approaches often rely on templated or manually crafted textual descriptions of traffic scenes, which lack the richness and fidelity needed to capture complex, real-world dynamics, leading to significant information loss, especially in visually grounded or ambiguous situations. Inference speed poses another hurdle: multi-step reasoning typically makes LLMs orders of magnitude slower than RL controllers, rendering LLMs impractical as standalone controllers in traffic environments where both latency and precision are essential. Moreover, current TSC simulators, such as the widely used SUMO [26] and CityFlow [27], can only provide statistical data for traffic intersections and cannot render real-time images, limiting the possibility to fuse the visual domain with language-based reasoning for traffic signal control.

To address the limitations of both RL- and LLM-based methods, we propose **VLMLight**, a safety-critical traffic signal control framework that integrates vision-language meta-control with dual-branch reasoning. At the core of VLMLight is a novel, custom-built traffic simulator that, for the first time, enables multiview image inputs at urban intersections, allowing policies to reason over rich visual cues such as vehicle types, spatial density, and dynamic motion patterns. Each intersection scene is first processed by VLM to produce interpretable, multi-directional traffic descriptions. These structured scene summaries are then passed to LLM acting as a meta-controller, which determines whether the situation requires low-latency control or high-level reasoning based on semantic understanding of the junction status. For routine traffic, VLMLight invokes a pre-trained RL policy to ensure fast and efficient signal control. In contrast, when a safety-critical condition is detected (e.g., an ambulance requiring priority passage), the system activates a slow deliberative reasoning branch in which multiple specialized LLM agents collaborate through structured dialogue. These agents sequentially perform phase analysis, action planning, and rule compliance verification, simulating a human-like deliberation process. This architecture fuses the speed of learned policies with the interpretability and robustness of language-based reasoning. Empirical results show that VLMLight reduces emergency vehicle waiting time by up to **65%** compared to RL-only baselines. Meanwhile, it maintains comparable performance in standard traffic with less than a **1%** degradation, demonstrating strong potential for trustworthy and scalable deployment.

The key contributions of this work are as follows:
1. We propose VLMLight, a novel traffic signal control framework that integrates vision-language scene understanding for both routine traffic and safety-critical scenarios. The framework leverages the first image-based simulator supporting multi-view visual inputs at intersections, enabling real-time, context-aware decision-making and enhancing flexibility beyond traditional handcrafted traffic states.
2. Guided by an LLM-based meta-controller, VLMLight dynamically selects between a fast RL policy and a structured reasoning module based on real-time junction context. This dual-branch architecture ensures both high efficiency in standard traffic and deliberate, high-level reasoning for safety-critical events, such as prioritizing emergency vehicles.
3. Experimental results show that VLMLight reduces the waiting time for emergency vehicles (e.g., ambulances) by up to **65%** compared to RL-only baselines, while maintaining comparable performance in standard traffic with less than **1%** degradation.

## 2  Preliminaries

### 2.1  Traffic Signal Terminology

Figure 1 illustrates a typical four-way intersection. Each intersection comprises two types of approaches: incoming approach with a varying number of income lanes ($l_{in}$), which carries traffic into the intersection, and outgoing approach with a varying number of outgoing lanes ($l_{out}$), on which the vehicles can leave the intersection. A *movement*, denoted as $m$, refers to vehicles going from an incoming approach to an outgoing one. For example, movement $m_1$ includes $l_{in}^2$ and $l_{in}^3$. To regulate traffic flow safely, movements are grouped into *phases*, each representing a set of non-conflicting movements that receive a green light simultaneously. For instance, Phase 1 is defined as $p_1 = \{m_1, m_2\}$. The complete set of feasible signal phases is denoted as $\mathcal{P} = \{p_1, p_2, p_3, p_4\}$.

### 2.2  Vision-Based TSC Simulator

To overcome the limitations of conventional traffic simulators, we introduce the first vision-enabled traffic signal control simulator that supports multi-view visual inputs at intersections. As illustrated in Figure 1, the simulator allows configurable camera placement to replicate real-world monitoring setups, including a bird's-eye view for monitoring overall traffic flow (left) and directional views simulating roadside perspectives from each approach (right).

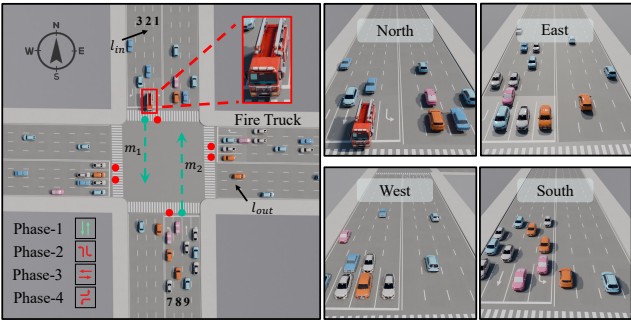

Figure 1: Illustration of a four-way intersection with four signal phases. The simulator supports multi-view visual inputs, including a bird's-eye view (left) and directional views from each approach (right), enabling lane-level observation of vehicle movements. In this example, the North-facing camera captures a fire truck traversing the intersection, highlighting the simulator's ability to support safety-critical reasoning through perceptually grounded traffic understanding.

These views form the perceptual basis for VLMLight. For example, the North-facing camera clearly captures a fire truck navigating through the intersection, and the lane markings provide additional guidance for tracking its trajectory. These visual cues support fine-grained scene understanding, which is essential for adaptive and safety-aware traffic signal control. By enabling real-time observation of dynamic intersection behavior in complex and realistic traffic scenarios, the simulator enhances the system's ability to make informed and context-aware signal decisions under both routine and high-stakes scenarios.

## 3  Method

We propose **VLMLight**, a vision-language traffic signal control framework that integrates the fast decision-making capabilities of RL with the generalization and structured reasoning strengths of LLMs. The core insight behind VLMLight is that routine traffic scenarios can be efficiently managed by a pre-trained RL policy. In contrast, rare or safety-critical situations, such as the presence of emergency vehicles, require interpretable and high-level reasoning that goes beyond the RL training distribution. As shown in Figure 2, VLMLight consists of four stages: **(1) Scene Understanding**, where a VLM processes multi-view intersection images into natural language descriptions; **(2) Safety-Prioritized Meta-Control**, where an LLM agent analyzes the scene and determines whether to activate the fast RL branch (for routine control) or the deliberative LLM reasoning branch (for complex or critical events); **(3) Routine Control**, where a pre-trained RL policy selects traffic signal actions based on spatio-temporal features; and **(4) Deliberative Reasoning**, where multiple LLM agents collaborate through structured dialogue to assess traffic priorities and verify rule compliance. By expressing perception, reasoning, and control processes in natural language, VLMLight provides a transparent interface that facilitates interpretability and post-hoc analysis across decision stages, thereby bridging low-level control and high-level reasoning in a unified framework. This hybrid

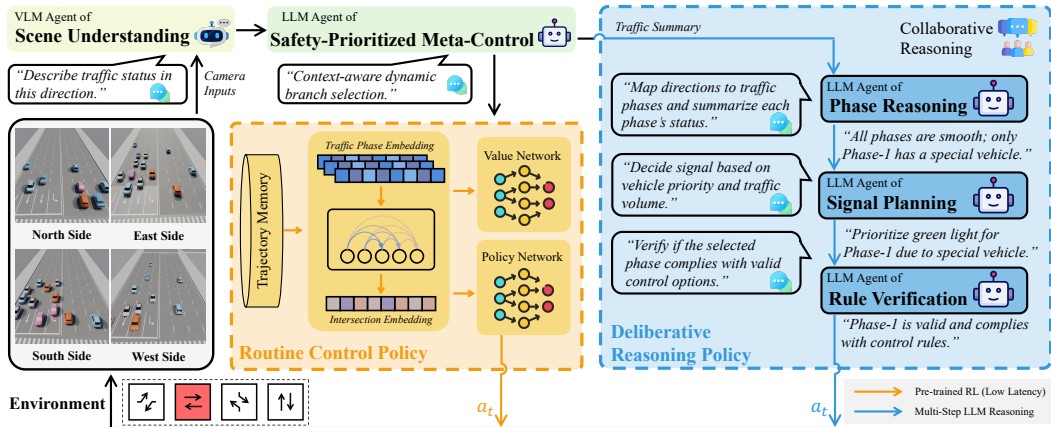

Figure 2: VLMLight architecture. Multi-view intersection images are first parsed by a VLM agent for scene understanding, after which a safety-prioritized LLM meta-controller interprets the scene and selects either a fast RL policy (orange) for routine traffic flow or a collaborative reasoning policy (blue) for safety-critical scenarios. A team of LLM agents—Phase Reasoning, Signal Planning, and Rule Verification—sequentially assess traffic phases, vehicle priority, and rule compliance to determine the final action $a_t$, giving real-time control and robust handling of complex events.

architecture enables VLMLight to maintain real-time responsiveness in typical scenarios while ensuring trustworthy and explainable decisions under safety-critical conditions.

## 3.1 Scene Understanding via VLM

Effective TSC requires an accurate and interpretable understanding of intersection conditions. Prior LLM-based approaches often rely on fixed, templated descriptions of traffic state (e.g., queue lengths or average delays), which are limited in expressiveness and fail to capture rich visual semantics, such as vehicle types, spatial configurations, or the presence of special vehicles. To address this, we build a closed-loop traffic simulator that provides real-time image observations from multiple directions of an intersection for perceptual grounding. Based on these visual inputs, a dedicated *Scene Understanding Agent* ($\texttt{Agent}_{\text{Scene}}$) leverages a pretrained VLM to process these inputs and generate natural language traffic scene descriptions.

Given directional images $\{I_1, I_2, \ldots, I_D\}$ from $D$ camera viewpoints (e.g., North, East, South, West), the agent produces a set of textual summaries:

$$\{T_1, T_2, \ldots, T_D\} = \texttt{Agent}_{\text{Scene}}(\{I_1, I_2, \ldots, I_D\}), \tag{1}$$

where each $T_i$ is a textual description of the traffic conditions in direction $i$, including lane-level semantics, congestion level, and whether any emergency or special vehicles (e.g., ambulances) are present. These free-form yet structured descriptions serve as interpretable inputs for downstream decision modules. As illustrated in Figure 3, this process is demonstrated on a T-junction with three incoming directions. For each $I_i$, the $\texttt{Agent}_{\text{Scene}}$ generates a corresponding $T_i$ that captures actionable traffic semantics, forming the perceptual basis for mode selection and phase reasoning in later stages.

## 3.2 Safety-Prioritized Meta-Control

Once the directional scene descriptions $\{T_1, \ldots, T_D\}$ are generated, the system must determine the appropriate decision-making strategy. RL-based policies are efficient for routine traffic but are limited by fixed reward structures, making them unsuitable for unexpected or safety-critical events. In contrast, LLM-based reasoning offers greater flexibility but incurs high latency, which is impractical for real-time control.

To balance responsiveness with adaptability, we introduce a *Safety-Prioritized Meta-Controller*, implemented as a language model agent ($\texttt{Agent}_{\text{ModeSelector}}$). Rather than issuing control actions directly, $\texttt{Agent}_{\text{ModeSelector}}$ interprets the set of textual scene descriptions and decides whether to route

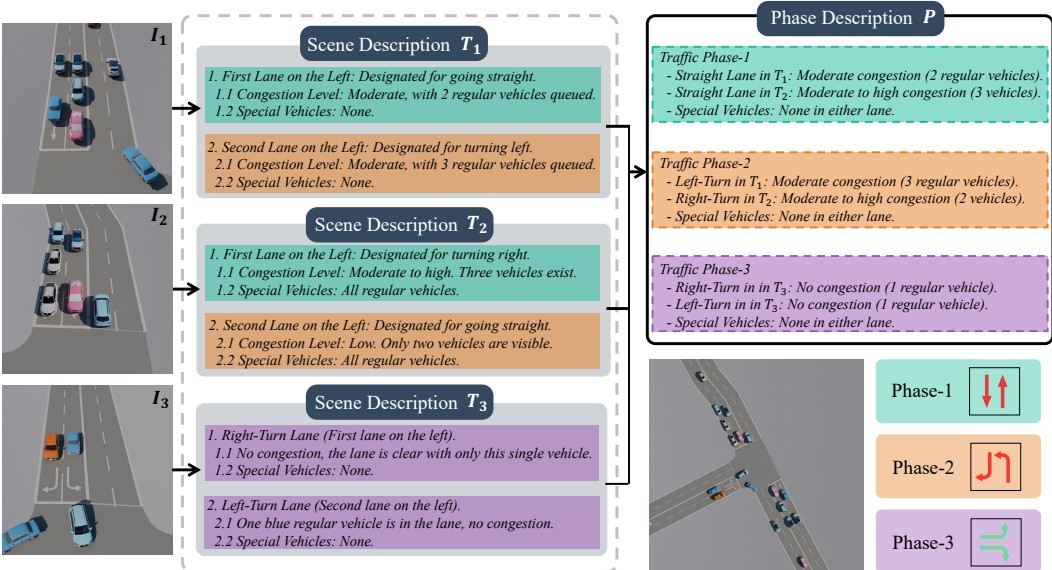

Figure 3: Illustration of Agent$_{\text{Scene}}$ in VLMLight. Given three-view images from a T-junction (left), a VLM-based Scene Description Agent generates directional-level textual summaries ($T_1$, $T_2$, $T_3$), describing lane semantics, congestion, and special vehicle presence. These summaries are then aggregated into phase-level descriptions ($P$) based on predefined signal phase mappings.

the system to the fast RL routine policy or the deliberative LLM reasoning module. If the scene is consistent with the assumptions of the RL training objective, such as the absence of special vehicles or traffic accidents, the system proceeds with low-latency control via the RL branch. However, if the description includes critical elements such as emergency vehicle detection, conflicting traffic goals, the meta-controller activates the structured reasoning path. This conditional routing allows VLMLight to respond efficiently in common scenarios while retaining the capability to reason explicitly under rare, high-stakes conditions.

## 3.3 Routine Control Policy

Under normal traffic conditions, VLMLight engages a fast RL decision branch designed for low-latency, high-throughput control. The current intersection state is constructed using recent statistics across up to 12 traffic movements (e.g., vehicle flow, occupancy, signal status), aggregated over the past five time steps to form a spatio-temporal input tensor.

This input is encoded by a lightweight Transformer-based module that first captures spatial dependencies among concurrent movements and then models temporal dynamics across frames. The resulting representation is used to select a traffic signal phase from a predefined discrete action set $\mathcal{A}$.

The RL policy is trained using Proximal Policy Optimization (PPO) [28], with a reward function designed to minimize intersection-level congestion and delay. This routine path enables fast, reactive control without invoking high-level reasoning, making it well-suited for the majority of non-critical traffic scenarios. Implementation details, including the agent design, state encoder, and PPO objective, are provided in Appendix A.1.

## 3.4 Deliberative Reasoning Policy

When traffic conditions deviate from routine patterns, such as the presence of emergency vehicles, conflicting priorities, or abnormal congestion, VLMLight activates its slow, deliberative reasoning branch. This path engages a team of specialized LLM agents that collaborate through structured natural language dialogue to generate a context-aware signal decision. The process unfolds in three stages: *Phase Reasoning*, *Signal Planning*, and *Rule Verification*.

**Phase Reasoning** The first step is to convert directional scene descriptions into phase-level traffic summaries aligned with the TSC control action space. As shown in Figure 3, given directional-level

descriptions $\{T_i\}$, the $\texttt{Agent}_{\text{Phase}}$ reorganizes the information using the predefined mapping from traffic movements to signal phases. For instance, Traffic Phase-1 will combine straight-going lanes from $T_1$ and $T_2$. The agent aggregates relevant semantic lane-level details (e.g., vehicle types, congestion levels) to form coherent descriptions $\{P_i\}$ for each candidate phase. This transformation bridges low-level visual perception and high-level traffic control, enabling the system to make decisions directly in the discrete space of traffic phases instead of raw directional inputs.

**Signal Planning**  Next, $\texttt{Agent}_{\text{Plan}}$ evaluates the candidate phase descriptions $\{P_i\}$ in light of the current control objectives, such as minimizing delay, prioritizing emergency vehicles, or balancing directional load. Based on this reasoning, it selects the most appropriate phase $a_t^{\text{LLM}} \in \mathcal{A}$ and generates a textual explanation justifying the decision. This rationale provides interpretability and enables auditability of the agent's decision-making process.

**Rule Verification**  Finally, $\texttt{Agent}_{\text{Check}}$ verifies whether the chosen action $a_t^{\text{LLM}}$ complies with the current feasible phase set $\mathcal{A}_t$. If the action is valid, the system proceeds with execution. If not, the agent selects an alternative from $\mathcal{A}_t$ that best aligns with the original intent while maintaining safety and consistency. The final decision is then formatted as a JSON object for downstream execution.

## 4  Experiments

### 4.1  Experimental Setup

**Experiment Settings**  All experiments are conducted in our custom-built TSC simulator, which integrates SUMO [26] to simulate vehicle dynamics and supports multi-view image rendering for vision-based perception. Each traffic episode adheres to standard urban signal timing, including a green phase (minimum duration of 10 s), a 3-second yellow phase, and a red phase. A minimum headway of 2.5 meters is enforced to reflect safe urban driving behavior. For vision-language processing, we use Qwen2.5-VL-32B [29] to generate structured traffic scene descriptions from multi-view images. In safety-critical scenarios, high-level reasoning is performed by Qwen2.5-72B [30] via structured multi-agent dialogue among three LLM agents. While Qwen is the primary backbone for our experiments, VLMLight is modular and compatible with alternative VLM and LLM models. We analyze the impact of model choices in Appendix C.2.

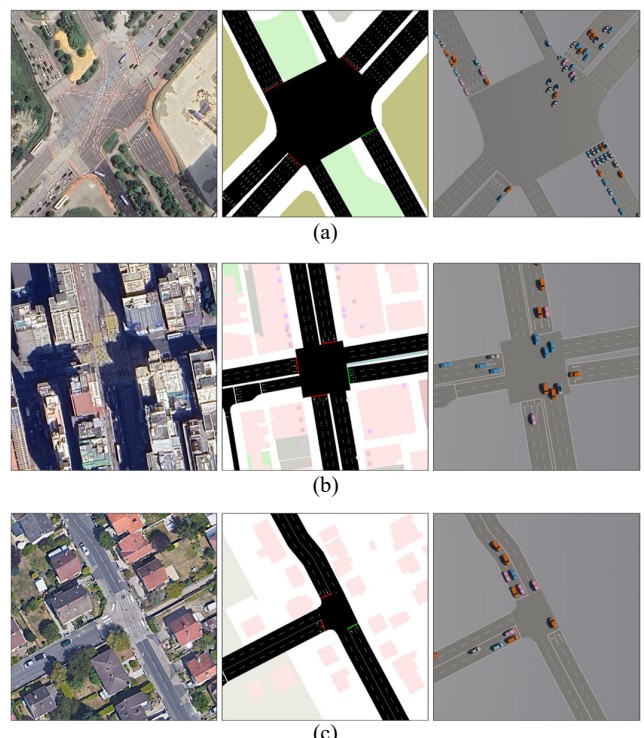

Figure 4: Three real-world intersections, each shown with three image modalities: (a) Songdo (South Korea), (b) Yau Ma Tei (Hong Kong), and (c) Massy (France). For each site, the satellite view is on the left, SUMO simulation in the middle, and our simulator rendering on the right.

**Dataset**  We evaluate VLMLight on traffic data collected from three real-world intersections located in Songdo (South Korea), Yau Ma Tei (Hong Kong), and Massy (France), as illustrated in Figure 4. These locations represent diverse urban settings. Songdo, a newly developed urban district, features larger intersections with up to five lanes per direction. Yau Ma Tei, situated in a dense urban core, has narrower roads and movement

restrictions (e.g., no left or right turns in certain directions). Massy provides a contrasting layout with a T-junction and distinct lane configurations. The diversity of intersection types—including crossroads (Songdo and Yau Ma Tei) and a T-junction (Massy)—as well as differences in size and movement patterns, enables more comprehensive evaluation of the method's generalizability across varied traffic scenarios. Each intersection is equipped with multi-directional cameras capturing 30 minutes of traffic data. The first 20 minutes are used for training the RL policy, while the remaining 10 minutes are reserved for testing. This setup allows us to evaluate the generalizability of VLMLight across varying topologies, lane geometries, and movement constraints. Additional dataset details are provided in Appendix B.2.

**Evaluation Metrics**    We evaluate TSC system performance using four metrics that jointly assess traffic efficiency and emergency vehicle handling. For overall traffic flow, we report the Average Travel Time (ATT), defined as the mean duration for vehicles to reach their destinations, and the Average Waiting Time (AWT), which represents the average duration that vehicles remain nearly stationary (speed < 0.1 m/s), typically due to signal-induced delays. To specifically evaluate emergency vehicle treatment, we measure Average Emergency Travel Time (AETT) and Average Emergency Waiting Time (AEWT). These reflect the system's responsiveness to high-priority traffic. Together, these metrics capture both global intersection efficiency and the framework's ability to prioritize safety-critical scenarios.

**Compared Methods**    To evaluate the performance of VLMLight, we compare it against both traditional and RL-based TSC methods. The traditional baselines include FixTime, Webster [3], and MaxPressure [4], which rely on handcrafted timing rules or pressure-based heuristics. For RL-based approaches, we consider IntelliLight [11], UniTSA [9], A-CATs [6], 3DQN-TSCC [7], and CCDA [10], which learn policies based on vectorized traffic state representations. Since VLMLight is the first framework that supports image-based input and leverages vision-language models to perform real-time traffic signal control, there are no existing VLM-based methods available for direct comparison. Therefore, to provide a more targeted comparison, we additionally include a baseline named Vanilla-VLM, which directly uses VLM-generated descriptions for decision-making without VLMLight's safety-critical meta-control and dual-branch reasoning architecture. A detailed description of these methods is provided in Appendix B.3.

## 4.2   Performance under Routine and Safety-Critical Traffic Scenarios

We evaluate VLMLight across three real-world intersection datasets and compare it with rule-based, RL-based, and VLM-based baselines. As shown in Table 1, VLMLight achieves strong performance in both routine and emergency scenarios, demonstrating a robust tradeoff between control efficiency and safety-aware decision-making.

**Routine Traffic Efficiency**    VLMLight maintains near-optimal performance under standard traffic conditions. Compared to the best-performing RL-based methods, the increase in ATT is marginal—less than 1% across all datasets. For example, in the Songdo intersection, ATT increases slightly from 86.80 s (UniTSA) to 87.14 s (VLMLight). Similarly, average waiting time (AWT) increases from 39.53 s to 39.73 s. These minor differences reflect the effectiveness of VLMLight's meta-controller, which defers to the fast RL branch in routine settings, avoiding unnecessary LLM overhead. By contrast, Vanilla-VLM, which relies solely on vision-language reasoning, performs significantly worse due to the semantic burden of integrating perception and decision-making within a single monolithic pipeline.

**Emergency Vehicle Prioritization**    In safety-critical scenarios, VLMLight activates its deliberative reasoning branch, leading to substantial improvements in emergency vehicle handling. Across all datasets, VLMLight reduces both AETT and AEWT by over 60% relative to RL-only baselines. For example, in the Songdo dataset, AEWT drops from 22.0 s to just 7.48 s. While Vanilla-VLM also benefits from explicit scene reasoning, its lack of structured phase mapping and decision modularity leads to frequent action errors, especially in complex intersections like Yau Ma Tei. This underscores the importance of decomposing the TSC task into modular reasoning steps. By assigning specialized roles to collaborating LLM agents' reasoning, VLMLight ensures correct, auditable actions and maintains real-time responsiveness even in safety-critical scenarios. Detailed analysis and examples of the intermediate reasoning steps generated by VLMLight are provided in Appendix D.

Table 1: Performance comparison on the three intersections. The top three results are marked with $*$ (best), $\dagger$ (second), and $\ddagger$ (third).

| Category | Method | South Korea, Songdo | | | |
|---|---|---|---|---|---|
| | | ATT ↓ | AWT ↓ | AETT ↓ | AEWT ↓ |
| Rule-based | FixTime | $111.73 \pm 5.11$ | $59.68 \pm 1.95$ | $108.68 \pm 7.20$ | $49.53 \pm 2.08$ |
| | Webster [3] | $102.89 \pm 4.20$ | $50.02 \pm 2.75$ | $82.77 \pm 3.94$ | $26.60 \pm 1.62$ |
| | MaxPressure [4] | $93.65 \pm 3.41$ | $43.71 \pm 1.61$ | $79.38 \pm 2.53$ | $35.38 \pm 2.42$ |
| RL-based | IntelliLight [11] | $\mathbf{87.12} \pm 5.10^{\dagger}$ | $\mathbf{39.68} \pm 1.98^{\dagger}$ | $\mathbf{70.00} \pm 2.36^{\ddagger}$ | $22.12 \pm 1.27$ |
| | UniTSA [9] | $\mathbf{86.80} \pm 4.89^{*}$ | $\mathbf{39.53} \pm 1.97^{*}$ | $69.74 \pm 3.80$ | $\mathbf{22.04} \pm 0.77^{\ddagger}$ |
| | A-CATs [6] | $88.23 \pm 6.01$ | $41.61 \pm 1.80$ | $70.08 \pm 4.40$ | $22.15 \pm 0.95$ |
| | 3DQN-TSCC [7] | $99.09 \pm 5.68$ | $45.13 \pm 2.20$ | $79.62 \pm 3.25$ | $25.17 \pm 1.54$ |
| | CCDA [10] | $89.32 \pm 6.21$ | $40.68 \pm 2.48$ | $71.76 \pm 4.82$ | $22.68 \pm 1.25$ |
| VLM-based | Vanilla-VLM | $105.48 \pm 17.28$ | $48.09 \pm 8.98$ | $\mathbf{60.38} \pm 11.78^{\dagger}$ | $\mathbf{11.05} \pm 1.73^{\dagger}$ |
| | **VLMLight (Ours)** | $\mathbf{87.14} \pm 4.98^{\ddagger}$ | $\mathbf{39.73} \pm 1.71^{\ddagger}$ | $\mathbf{49.88} \pm 2.42^{*}$ | $\mathbf{7.48} \pm 0.45^{*}$ |

| Category | Method | Hongkong, Yau Ma Tei | | | |
|---|---|---|---|---|---|
| | | ATT ↓ | AWT ↓ | AETT ↓ | AEWT ↓ |
| Rule-based | FixTime | $67.63 \pm 4.57$ | $40.00 \pm 2.28$ | $82.67 \pm 3.23$ | $53.17 \pm 2.14$ |
| | Webster [3] | $56.26 \pm 3.39$ | $28.62 \pm 1.23$ | $59.67 \pm 4.11$ | $30.83 \pm 1.79$ |
| | MaxPressure [4] | $41.36 \pm 2.22$ | $13.33 \pm 0.40$ | $36.17 \pm 2.39$ | $8.83 \pm 0.27$ |
| RL-based | IntelliLight [11] | $\mathbf{38.07} \pm 2.52^{*}$ | $\mathbf{10.28} \pm 0.54^{*}$ | $\mathbf{33.17} \pm 1.54^{\ddagger}$ | $\mathbf{5.17} \pm 0.18^{\ddagger}$ |
| | UniTSA [9] | $\mathbf{38.10} \pm 1.28^{\dagger}$ | $\mathbf{10.29} \pm 0.50^{*}$ | $33.19 \pm 1.92$ | $5.17 \pm 0.35$ |
| | A-CATs [6] | $42.64 \pm 1.43$ | $11.51 \pm 0.71$ | $37.14 \pm 1.67$ | $5.79 \pm 0.34$ |
| | 3DQN-TSCC [7] | $46.20 \pm 2.01$ | $12.47 \pm 0.54$ | $40.25 \pm 2.05$ | $6.27 \pm 0.32$ |
| | CCDA [10] | $41.60 \pm 2.02$ | $11.23 \pm 0.45$ | $36.24 \pm 1.45$ | $5.65 \pm 0.31$ |
| VLM-based | Vanilla-VLM | $62.45 \pm 8.76$ | $17.62 \pm 2.45$ | $\mathbf{27.79} \pm 3.79^{\dagger}$ | $\mathbf{5.86} \pm 0.69^{\dagger}$ |
| | **VLMLight (Ours)** | $\mathbf{39.80} \pm 1.65^{\ddagger}$ | $\mathbf{11.85} \pm 0.60^{\ddagger}$ | $\mathbf{13.50} \pm 0.56^{*}$ | $\mathbf{2.17} \pm 0.12^{*}$ |

| Category | Method | France, Massy | | | |
|---|---|---|---|---|---|
| | | ATT ↓ | AWT ↓ | AETT ↓ | AEWT ↓ |
| Rule-based | FixTime | $75.84 \pm 4.46$ | $28.19 \pm 1.58$ | $73.60 \pm 4.62$ | $27.80 \pm 1.06$ |
| | Webster [3] | $68.92 \pm 2.15$ | $20.89 \pm 0.79$ | $65.20 \pm 3.04$ | $19.80 \pm 0.61$ |
| | MaxPressure [4] | $64.82 \pm 4.01$ | $15.25 \pm 0.86$ | $72.40 \pm 2.83$ | $22.40 \pm 0.92$ |
| RL-based | IntelliLight [11] | $\mathbf{57.73} \pm 3.51^{*}$ | $\mathbf{9.80} \pm 0.67^{*}$ | $\mathbf{54.80} \pm 2.45^{\ddagger}$ | $\mathbf{7.81} \pm 0.39^{\ddagger}$ |
| | UniTSA [9] | $\mathbf{57.84} \pm 1.91^{\dagger}$ | $\mathbf{9.82} \pm 0.54^{\dagger}$ | $64.90 \pm 2.28$ | $9.81 \pm 0.41$ |
| | A-CATs [6] | $62.44 \pm 3.24$ | $10.60 \pm 0.35$ | $59.27 \pm 2.72$ | $7.35 \pm 0.30$ |
| | 3DQN-TSCC [7] | $69.10 \pm 2.86$ | $13.73 \pm 0.52$ | $65.59 \pm 2.80$ | $8.14 \pm 0.49$ |
| | CCDA [10] | $62.83 \pm 2.42$ | $11.19 \pm 0.38$ | $58.80 \pm 2.40$ | $7.20 \pm 0.49$ |
| VLM-based | Vanilla-VLM | $81.18 \pm 11.91$ | $13.44 \pm 1.91$ | $\mathbf{53.11} \pm 8.03^{\dagger}$ | $\mathbf{3.04} \pm 0.61^{\dagger}$ |
| | **VLMLight (Ours)** | $\mathbf{60.84} \pm 2.63^{\ddagger}$ | $\mathbf{11.49} \pm 0.75^{\ddagger}$ | $\mathbf{45.40} \pm 2.88^{*}$ | $\mathbf{2.60} \pm 0.10^{*}$ |

## 4.3 Ablation Study

We conduct ablation studies to quantify the impact of each agent module in the Deliberative Reasoning branch. As shown in Table 2, disabling $\text{Agent}_{\text{Phase}}$ leads to a notable increase in AEWT from 48s to 58s, indicating its essential role in the reasoning pipeline. This agent plays a crucial role by abstracting low-level directional features into structured traffic phase representations, enabling downstream reasoning to operate over a discrete and legally grounded action space. Without this abstraction, the reasoning agent must handle raw directional information directly, which leads to error-prone or inapplicable decisions in intersections with complex phase-movement mappings.

In contrast, removing $\text{Agent}_{\text{Check}}$, which verifies the legality of the final decision—has limited effect on overall performance, as invalid actions are rare in most cases. These results highlight the importance of structured semantic grounding and explicit planning in traffic signal control. VLMLight's

Table 2: Ablation results on Songdo, impact of structured reasoning components and corresponding inference time.

| Modules | | | Metrics | | |
|---|---|---|---|---|---|
| $\text{Agent}_{\text{Phase}}$ | $\text{Agent}_{\text{Plan}}$ | $\text{Agent}_{\text{Check}}$ | AETT $\downarrow$ | AEWT $\downarrow$ | Time ($\downarrow$, s) |
| ✗ | ✓ | ✗ | 58.93 | 9.03 | 7.10 |
| ✓ | ✓ | ✗ | 49.96 | 8.12 | 10.97 |
| ✓ | ✓ | ✓ | **48.88** | **7.48** | 11.48 |

modular agent design not only improves emergency response but also supports transparent, auditable decision-making in safety-critical environments.

We further evaluate the inference latency introduced by the structured reasoning branch under the Songdo intersection. As reported in the last column of Table 2, we compare different configurations of VLMLight modules to measure their computational overhead. Even with all modules enabled, the total decision latency remains 11.48 seconds. This latency is well within the acceptable range for real-time deployment, as it fits the typical signal phase buffer (10s green + 3s yellow). Detailed latency breakdowns and further analysis are provided in Appendix C.3.

## 5 Related Work

**Traffic Signal Control** TSC methods fall into three categories: rule-based, RL-based, and LLM-based. Traditional approaches like Webster's method [3], SOTL [5], and MaxPressure [4] use fixed heuristics and perform well under steady traffic, but struggle with real-time adaptability [2]. RL-based methods [6, 31, 9, 32, 33, 34, 13, 35] improve responsiveness by learning from interaction, but often overlook safety-critical events due to simplified states and static rewards [36, 37, 38, 39]. Recent LLM-based approaches [21, 23, 22, 25] introduce high-level reasoning for rare or long-tail cases, but rely on hand-crafted text inputs and lack true visual grounding. Moreover, they underperform in routine traffic control compared to specialized RL agents.

**Simulators for TSC** Most TSC simulators, such as SUMO [26], CityFlow [27], and LibSignal [40], provide only structured traffic states (e.g., vehicle counts, signal phases) without visual outputs, limiting their use in vision-grounded reasoning. High-fidelity driving simulators such as CARLA [41] and MetaDrive [42] offer photorealistic rendering, but they are primarily designed for autonomous driving tasks rather than intersection control, and they often lack native support for signal scheduling or require extensive customization. To address this gap, SynTraC [43] introduces an image-based dataset for TSC built upon CARLA, providing intersection images annotated with traffic states, signal phases, and reward information under diverse weather and lighting conditions. Although SynTraC represents a significant step toward visual perception in TSC, it remains limited to single-intersection scenarios and does not support user-defined or multi-view configurations, constraining its scalability and applicability to broader traffic management research.

## 6 Conclusion

We present **VLMLight**, a vision-language TSC framework that unifies fast policy execution with structured semantic reasoning. By dynamically selecting between a reinforcement learning policy for routine traffic and a deliberative reasoning branch for safety-critical cases, VLMLight adapts to diverse intersection scenarios with both efficiency and robustness. Empirical experiment results demonstrate that our method reduces emergency vehicle waiting time by up to **65%**, while maintaining comparable performance to RL-only baselines in standard conditions. Importantly, the full inference process completes within 11.5 seconds, which fits comfortably within the typical 13 s signal phase buffer used in urban deployments, indicating practical feasibility for real-time use. In addition, we release the first vision-based TSC simulator with support for multi-view image inputs and dynamic scene rendering. This simulator enables richer visual understanding of intersection conditions and provides a foundation for future research into perceptually grounded and interpretable traffic control strategies.

**Limitation** VLMLight has two primary limitations. First, our simulator currently lacks diverse weather and lighting conditions, limiting the evaluation of visual robustness under challenging environments. Second, all experiments are conducted on single intersections. Extending the framework to multi-intersection settings requires further study on the scalability and coordination of vision-language reasoning across a broader traffic network.

**Broader Impacts** VLMLight provides a vision-language framework for TSC that enhances safety and efficiency through real-time visual reasoning. The open-source simulator offers a valuable tool for future research on perception-based traffic systems.

## Acknowledgments and Disclosure of Funding

This work was supported in part by the Guangdong Science and Technology Department under Grant 2025SF0001.

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

# Appendix

In this appendix, we provide supplementary material to further elaborate on VLMLight:

- Additional method details, including the routine control policy, full algorithm, and prompt templates in Section A.
- Full experimental setup, including datasets and compared baselines in Section B.
- Extended results, including RL convergence curves, LLM model comparisons, and inference times in Section C.
- Three case studies illustrating the decision-making process of VLMLight in Section D.
- In-depth discussion on limitations and broader impact in Section E.

## A  Method

### A.1  Details of Routine Control Policy

In standard traffic scenarios, VLMLight adopts an Reinforcement Learning (RL) policy trained within a Markov Decision Process (MDP) framework. At each timestep $t$, the intersection state is encoded as $J_t = [\mathbf{m}_1^t, \ldots, \mathbf{m}_{12}^t]$, where each $\mathbf{m}_i^t \in \mathbb{R}^7$ represents the status of the $i$-th movement at the intersection. The vector $\mathbf{m}_i^t$ includes a combination of traffic flow, movement, and signal-related features. The traffic characteristics are captured through the average vehicle flow $F^{i,t}$, maximum occupancy $O_{\max}^{i,t}$, and mean occupancy $O_{\text{mean}}^{i,t}$ since the last control action. The movement-specific features consist of an indicator $I_s^i \in \{0, 1, 2\}$ that reflects whether the movement is straight, left, or right, and the lane count $L_i$. Signal status is described by two binary indicators: $I_{\text{cg}}^{i,t}$ for whether the current phase is green, and $I_{\text{mg}}^{i,t}$ to indicate whether the minimum green duration requirement has been satisfied. The complete feature vector is written as:

$$\mathbf{m}_i^t = \left[ F^{i,t}, O_{\max}^{i,t}, O_{\text{mean}}^{i,t}, I_s^i, L_i, I_{\text{cg}}^{i,t}, I_{\text{mg}}^{i,t} \right]. \tag{2}$$

To ensure a consistent input size, intersections with fewer than 12 movements are zero-padded. For example, at the Yau Ma Tei intersection, the absence of a left-turn movement from north to south is represented by a zero vector. An illustration of the zero-padding scheme is shown in Figure 5. To capture temporal dynamics, the agent receives input from the current and previous four timesteps, forming a 5-frame observation window:

$$S_t = [J_{t-4}, J_{t-3}, J_{t-2}, J_{t-1}, J_t] \in \mathbb{R}^{5 \times 12 \times 7}. \tag{3}$$

At each timestep, the agent selects an action $a_t \in \mathcal{P}$, where $\mathcal{P}$ denotes the set of available traffic signal phases. The reward $r_t$ is defined as the negative average queue length, encouraging smoother traffic flow.

To extract expressive representations from $S_t$, we use a Transformer-based encoder that processes both spatial and temporal dimensions. In the spatial encoding stage, the individual movement vectors $\mathbf{m}_i^t \in \mathbb{R}^7$ for each frame $J_t \in \mathbb{R}^{12 \times 7}$ are projected to $d$-dimensional embeddings $\mathbf{h}_i^t$ via a shared linear layer:

$$\mathbf{h}_i^t = \mathbf{W}_e \mathbf{m}_i^t + \mathbf{b}_e, \quad i = 1, \ldots, 12, \tag{4}$$

producing a token matrix $\mathbf{h}^t = [\mathbf{h}_1^t, \ldots, \mathbf{h}_{12}^t] \in \mathbb{R}^{12 \times d}$. This matrix is processed by a Transformer block composed of layer normalization (LN), multihead self-attention (MSA), and a feed-forward network (MLP):

$$\mathbf{E}^t = \text{MLP}\left(\text{LN}\left(\mathbf{h}^t + \text{MSA}(\text{LN}(\mathbf{h}^t))\right)\right) \in \mathbb{R}^{12 \times d}. \tag{5}$$

We then apply mean pooling across the 12 movement embeddings to obtain a compact spatial summary $\mathbf{s}_t$:

$$\mathbf{s}_t = \frac{1}{12} \sum_{i=1}^{12} \mathbf{E}_i^t \in \mathbb{R}^d. \tag{6}$$

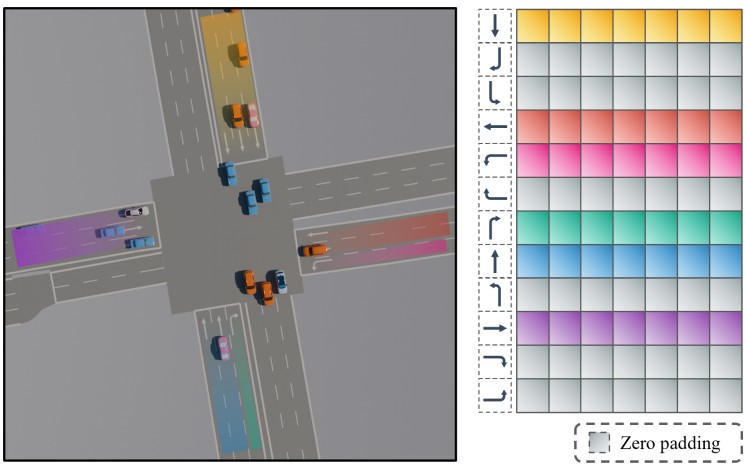

Figure 5: Example of zero-padding at the Yau Ma Tei intersection.

In the second stage, we process the five temporal embeddings $\mathbf{S}_t = [\mathbf{s}_{t-4}, \ldots, \mathbf{s}_t] \in \mathbb{R}^{5 \times d}$ using another Transformer encoder:

$$\mathbf{Z}_t = \text{MLP} \left( \text{LN} \left( \mathbf{S}_t + \text{MSA} \left( \text{LN}(\mathbf{S}_t) \right) \right) \right) \in \mathbb{R}^{5 \times d}. \tag{7}$$

where each row $\mathbf{Z}_{t,k} \in \mathbb{R}^d$ corresponds to the enriched representation of time step $t - k$. To obtain a fixed-length state embedding for decision-making, we apply average pooling over the temporal axis:

$$\mathbf{f}_t = \frac{1}{5} \sum_{k=1}^{5} \mathbf{Z}_{t,k} \in \mathbb{R}^d. \tag{8}$$

After obtaining the spatio-temporal representation $\mathbf{h}_t$, we pass it into both the policy and value networks. The policy head outputs a probability distribution $\pi(a_t \mid \mathbf{h}_t; \theta)$ over actions, and the value head estimates the expected return $V(\mathbf{h}_t; \phi)$. We optimize the policy using the Proximal Policy Optimization (PPO) algorithm, which maximizes the following surrogate objective:

$$\mathcal{L}^{\text{PPO}}(\theta) = \mathbb{E}_t \left[ \min \left( r_t(\theta) \hat{A}_t, \text{clip}(r_t(\theta), 1 - \epsilon, 1 + \epsilon) \hat{A}_t \right) \right], \tag{9}$$

where $r_t(\theta) = \frac{\pi_\theta(a_t \mid \mathbf{h}_t)}{\pi_{\theta_{\text{old}}}(a_t \mid \mathbf{h}_t)}$ is the policy ratio and $\hat{A}_t$ is the advantage estimate. The clipping parameter $\epsilon$ is used to bound policy updates and improve training stability. The value network is trained by minimizing the squared error between predicted values and empirical returns:

$$\mathcal{L}^{\text{value}}(\phi) = \mathbb{E}_t \left[ \left( V(\mathbf{h}_t; \phi) - \hat{R}_t \right)^2 \right], \tag{10}$$

where $\hat{R}_t$ denotes the bootstrap return. This hierarchical design enables the routine control policy to leverage both short-term dynamics and long-term patterns for efficient traffic management. The full training objective combines the above terms:

$$\mathcal{L}_{\text{total}}(\theta, \phi) = -\mathcal{L}_{\text{policy}}(\theta) + \lambda_v \mathcal{L}_{\text{value}}(\phi), \tag{11}$$

where $\lambda_v$ is a hyperparameter that balances value learning. This Transformer-based hierarchical design allows the fast RL policy to effectively reason over fine-grained spatio-temporal signals for efficient traffic control in routine scenarios.

## A.2 Algorithm for VLMLight

VLMLight employs a modular set of collaborative agents that together enable perception-aware, safety-critical traffic control. Table 3 summarizes the responsibilities of each agent. The architecture is designed to interleave fast decision-making (via RL) with high-level reasoning (via LLM agents)

Table 3: Summary of agent roles in VLMLight.

| Agent Name | Function |
|---|---|
| $\text{Agent}_{\text{Scene}}$ | Converts multi-view images $I_i$ into directional text descriptions $T_i$ |
| $\text{Agent}_{\text{ModeSelector}}$ | Selects control mode: fast RL policy or structured LLM reasoning |
| $\text{Agent}_{\text{Phase}}$ | Aggregates $T_i$ into phase-level descriptions $P_i$ |
| $\text{Agent}_{\text{Plan}}$ | Selects optimal action $a_t^{\text{LLM}}$ and explains the rationale |
| $\text{Agent}_{\text{Check}}$ | Validates action feasibility against current legal phase set $\mathcal{A}$ |

under a unified meta-control mechanism. Each agent operates on structured inputs—either visual, textual, or phase-level representations—and outputs either a decision or an intermediate semantic representation used by downstream agents.

---

**Algorithm 1** Algorithm for VLMLight

---

**Require:** Maximum simulation time $T_{\max}$, legal phase set $\mathcal{A}$, phase-to-lane mapping $\mathcal{M}_{\text{ph-lane}}$, maximum check attempts $N_{\text{check}}$, control interval $\Delta t$.
1: Initialize $t \leftarrow 0$
2: **while** $t < T_{\max}$ **do**
3:     Obtain multi-view images $\{I_1, I_2, \ldots, I_D\}$ from simulator
4:     $\{T_1, T_2, \ldots, T_D\} \leftarrow \text{AGENT}_{\text{SCENE}}(\{I_i\})$
5:     $m \leftarrow \text{AGENT}_{\text{MODESELECTOR}}(\{T_i\})$
6:     **if** $m = \text{RL}$ **then**
7:         $a_t \leftarrow \text{AGENT}_{\text{RL}}(s_t)$
8:     **else**
9:         $\{P_1, P_2, \ldots, P_K\} \leftarrow \text{AGENT}_{\text{PHASE}}(\{T_i\}, \mathcal{M}_{\text{ph-lane}})$
10:         $a_t^{\text{LLM}} \leftarrow \text{AGENT}_{\text{PLAN}}(\{P_k\})$
11:         **for** $n = 1$ **to** $N_{\text{check}}$ **do**
12:             $a_t \leftarrow \text{AGENT}_{\text{CHECK}}(a_t^{\text{LLM}}, \mathcal{A}_t)$
13:             **if** $a_t \in \mathcal{A}_t$ **then**
14:                 **break**
15:             **end if**
16:         **end for**
17:         **if** $a_t \notin \mathcal{A}_t$ **then**
18:             $a_t \leftarrow \text{AGENT}_{\text{RL}}(s_t)$
19:         **end if**
20:     **end if**
21:     Execute $a_t$ in simulator
22:     $t \leftarrow t + \Delta t$
23: **end while**

---

Algorithm 1 outlines the inference procedure of VLMLight. At each decision interval, the system receives multi-view images from the simulator and invokes the $\text{Agent}_{\text{Scene}}$ to generate directional scene descriptions. A safety-prioritized meta-controller $\text{Agent}_{\text{ModeSelector}}$ then determines whether to proceed with the fast RL policy or activate the structured reasoning branch. In routine conditions, a lightweight RL agent issues a control action based on the current traffic state. In contrast, for safety-critical scenarios, three LLM agents collaborate sequentially: the $\text{Agent}_{\text{Phase}}$ module transforms scene descriptions into phase-level summaries using a predefined phase-to-lane mapping $\mathcal{M}$ph-lane; the $\text{Agent}_{\text{Plan}}$ agent proposes a candidate action aligned with system objectives; and the $\text{Agent}_{\text{Check}}$ agent verifies the action's legality against the current feasible phase set $\mathcal{A}_t$. If the verification fails after $N_{\text{check}}$ attempts, the system falls back to the action proposed by $\text{Agent}_{\text{RL}}$ to ensure continued operation. This strict validation pipeline, coupled with the fallback mechanism, safeguards against potential noise or reasoning errors, ensuring that only feasible and safety-compliant actions are executed. The selected action is then executed in the simulator, and the simulation clock advances by a fixed interval $\Delta t$. This loop continues until the maximum simulation time $T$max is reached.

## A.3 Prompt Templates

This section presents the prompt templates used by the five agents in VLMLight. $\text{Agent}_{\text{Scene}}$ uses a VLM to convert intersection images into textual descriptions, as shown in Figure 6. The other four agents are based on LLMs: $\text{Agent}_{\text{ModeSelector}}$ determines the control mode (Figure 7), $\text{Agent}_{\text{Phase}}$ generates phase-level descriptions based on the scene context (Figure 8), $\text{Agent}_{\text{Plan}}$ selects the optimal phase (Figure 9), and $\text{Agent}_{\text{Check}}$ verifies rule compliance (Figure 10).

---

**$Agent_{Scene}$**

You are TrafficVision, an AI traffic analyst. Based on the intersection image below from a fixed surveillance camera, provide an accurate and concise description of the scene:
- Assess traffic congestion level.
- Identify special vehicles (e.g., ambulances, police cars, fire trucks) only if clearly visible.
- Avoid speculation — report only what is verifiable.
[{Image}]

Figure 6: Prompt template for $\text{Agent}_{\text{Scene}}$.

---

**$Agent_{ModeSelector}$**

You are in a role play game. The following roles are available:
- Routine Control Agent: Handle normal traffic using RL-based decisions to optimize flow and ensure safety.
- Reasoning Agent: Take over when special vehicles appear or unusual conditions arise, ensuring their priority while keeping traffic orderly. Please read the dialogue history and choose the next suitable role to speak.
When the user indicates to stop chatting or when the topic should be terminated, please return '[STOP]'.
Only return the role name from [{agent_names}] or '[STOP]'. Do not reply any other content.

Figure 7: Prompt template for $\text{Agent}_{\text{ModeSelector}}$.

---

**$Agent_{Phase}$**

You are a traffic phase analyst. The intersection has [{direction_number}] directional descriptions, each representing a different view. The following is the phase-to-lane mapping for this junction:
[{phase-to-lane}]
Please summarize each traffic phase by extracting:
1) Congestion level
2) Confirmed special vehicles (ambulance, police car, fire truck — only if clearly visible)
3) Any notable traffic events
Use the scene descriptions provided for each direction:
[{junction_description-direction}], …

Figure 8: Prompt template for $\text{Agent}_{\text{Phase}}$.

---

**$Agent_{Plan}$**

You are roleplaying as a traffic police officer managing a real-time intersection. You have received the description for each traffic phase: [{phase_description}].
Please make decisions by:
- Prioritizing confirmed emergency vehicles (ambulances, police cars, or fire trucks)
- Otherwise, adjusting signal timings to minimize congestion and maximize overall traffic efficiency
Note: You must choose only from the following available actions: [{available_actions}]

Figure 9: Prompt template for $\text{Agent}_{\text{Plan}}$.

You are an evaluator responsible for verifying whether a traffic control decision is valid.
A valid decision must select an action strictly from the following set: [{self.available_actions}].
If the decision is compliant, return a JSON object with exactly two keys:
- "decision": the selected Traffic Phase ID
- "explanation": the rationale for the decision based on the traffic phase description
No other keys are allowed in the output. Example output:
{
  "decision": "Phase-2",
  "explanation": "The image depicts a basic road intersection scenario with no special vehicle markings; …
}

Figure 10: Prompt template for $\text{Agent}_{\text{Check}}$.

## B  Experiments Setup

### B.1  Experiment Settings

In this section, we describe the experimental setup used to evaluate the performance of VLMLight, our proposed traffic signal control framework. The experiment involves the integration of a self-developed traffic simulator, the deployment of large models for vision-language understanding, and the training of an RL policy to assess VLMLight's adaptability in dynamic traffic scenarios, especially in safety-critical situations.

The experiments are conducted using a self-developed traffic simulation environment built on top of the SUMO (Version 1.22) framework. The simulated roads feature a maximum speed limit of 13.9 m/s (approximately 50 km/h) to model typical urban traffic conditions. The simulator includes three specialized types of vehicles—police cars, ambulances, and fire trucks—to evaluate the system's performance in emergency response scenarios. The vehicle speed distribution is modeled as a Gaussian distribution with a mean of 10 m/s and a variance of 3, reflecting typical urban traffic flow patterns.

To ensure high-performance traffic signal control, we deploy both VLMs and LLMs locally on a high-performance computing system. The system is equipped with an Intel Xeon 6738P CPU, 256 GB of RAM, and five A100 GPUs, all running Ubuntu 20.04 LTS. This setup allows us to run multiple VLMs in parallel, significantly improving the speed and efficiency of scene understanding. The use of multiple VLMs is essential for rapidly processing the visual inputs from various intersection views and generating natural language descriptions that capture the complex dynamics of the scene. This configuration ensures that VLMLight can make real-time decisions based on a thorough understanding of the current traffic situation.

For the RL-based components of VLMLight, we use the PPO [28] algorithm, implemented via the Stable Baselines3 library. To speed up training and improve the exploration of different traffic conditions, we deploy 30 parallel processes, each interacting with a separate instance of the simulation. The total number of environment steps is set to $3e5$ and the batch size is configured to $64$. The learning rate follows a linear schedule, starting at $1e-3$ and gradually decreasing as the number of training steps increases. Additionally, the trajectory memory size is set to 3000.

### B.2  Dataset Details

To evaluate the generalizability of VLMLight, we construct an evaluation suite based on three real-world intersections with varying topologies and traffic conditions, located in Songdo (South Korea), Yau Ma Tei (Hong Kong), and Massy (France). Each site was selected to represent distinct urban forms: Songdo features large-scale grid intersections with high traffic throughput; Yau Ma Tei is situated in a densely populated downtown with constrained geometry and restricted turning rules; and Massy contains a suburban T-junction with lighter traffic and fewer lanes. These variations allow us to systematically test VLMLight across a broad range of physical layouts and flow intensities.

For each intersection, multi-directional cameras were deployed to capture 30 minutes of continuous traffic footage, with all approaches covered. As shown in Figure 11, the first column in each subfigure

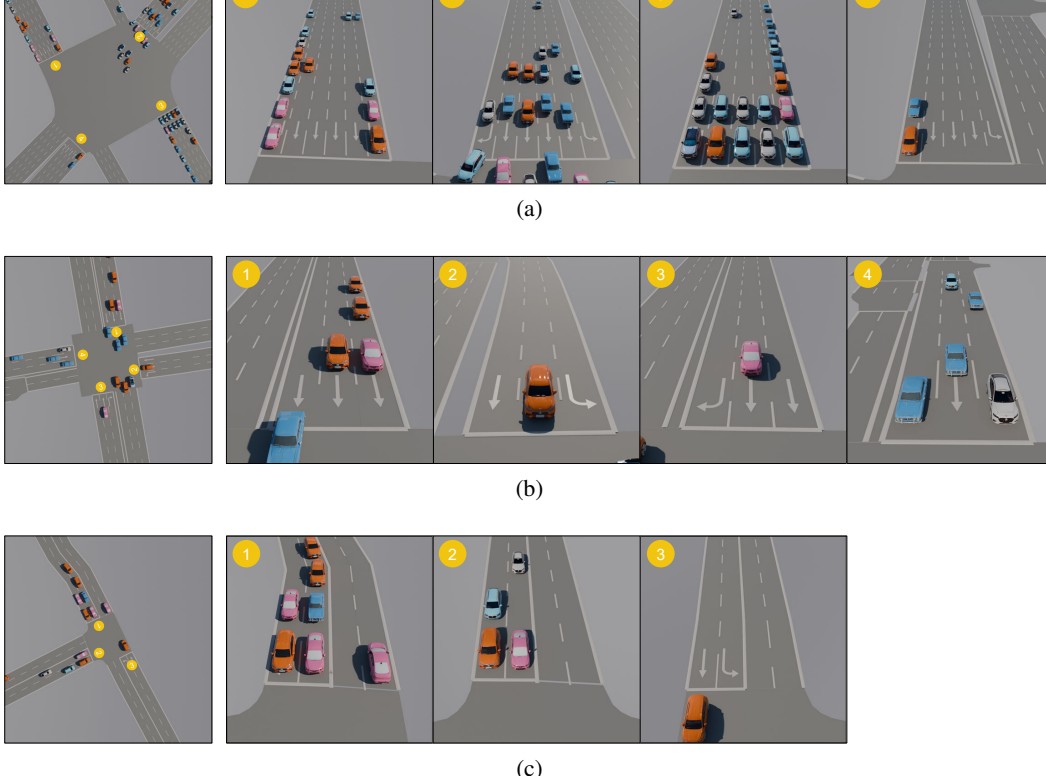

Figure 11: Multi-view camera observations of three real-world intersections. Top-down layouts are shown on the left; directional inbound views follow. (a) Songdo, (b) Yau Ma Tei, and (c) Massy.

presents the top-down intersection layout, while subsequent images show direction-specific views from each inbound lane. These direction-wise visual inputs are fed into the VLMLight perception module for downstream reasoning. Table 4 summarizes the directional traffic flow statistics. Vehicle counts and arrival rates vary considerably across sites: Songdo shows the heaviest traffic load, with arrival rates exceeding 2 vehicles/s in certain directions, while Massy represents the lightest scenario with sub-1 vehicle/s flow. Emergency vehicles were sparsely but consistently present across all sites, ensuring meaningful evaluation of safety-critical reasoning. This comprehensive setup enables reproducible and diverse benchmarking for vision-language-based traffic signal control.

### B.3 Compared Methods

In this section, we introduce the methods compared to our VLMLight framework, which includes three traditional baselines, five RL-based approaches, and one VLM-based method. These methods are evaluated to highlight the advantages of VLMLight in terms of traffic efficiency and safety.

**Traditional Methods.** We adopt three traditional approaches in experiments as follows:

- FixTime: Fixed-time control assigns predetermined cycle and phase durations, which are most effective in steady traffic conditions. We consider the FixTime-30 variant, where each phase duration is fixed at 30 seconds.
- Webster [3]: The Webster method adjusts cycle lengths and phase splits based on traffic volumes, optimizing travel time in uniform traffic. In this study, we use it for adjusting traffic lights based on real-time traffic flow.
- MaxPressure [4]: The MaxPressure method prioritizes phases with the highest traffic demand, optimizing the flow by minimizing congestion. This approach is known for its simplicity and effectiveness in maximizing intersection throughput.

Table 4: Traffic flow statistics for each approach direction at the three intersections. #Veh: total vehicles; #Emerg: emergency vehicles.

| Network | Dir | #Veh | #Emerg | Arrival Rate (vehicles/s) | | | |
|---|---|---|---|---|---|---|---|
| | | | | Mean | Std | Min | Max |
| Songdo | ① | 3780 | 9 | 2.10 | 0.31 | 1.60 | 2.67 |
| | ② | 3740 | 4 | 2.08 | 0.32 | 1.58 | 2.78 |
| | ③ | 2993 | 4 | 1.66 | 0.23 | 1.20 | 2.20 |
| | ④ | 2932 | 5 | 1.63 | 0.21 | 1.35 | 2.03 |
| Yau Ma Tei | ① | 2556 | 7 | 1.42 | 0.20 | 1.05 | 1.70 |
| | ② | 1916 | 4 | 1.06 | 0.17 | 0.78 | 1.37 |
| | ③ | 1927 | 4 | 1.07 | 0.17 | 0.75 | 1.35 |
| | ④ | 2346 | 2 | 1.30 | 0.20 | 1.00 | 1.65 |
| Massy | ① | 1216 | 3 | 0.68 | 0.09 | 0.45 | 0.85 |
| | ② | 626 | 2 | 0.35 | 0.06 | 0.25 | 0.47 |
| | ③ | 1079 | 2 | 0.60 | 0.10 | 0.42 | 0.78 |

**RL-Based Methods.** We examine five RL-based methods, each offering distinct strategies for TSC:

- IntelliLight [11]: IntelliLight uses a DQN-based approach to select the best traffic phase from available options, addressing data imbalance by maintaining a balanced data buffer for each phase. In this study, decisions are made every 5 s from all available phases.

- UniTSA [9]: UniTSA introduces junction matrices, which enable it to adapt to different intersection layouts. The method also leverages state augmentation, ensuring the agent encounters diverse intersection types and traffic volume during training.

- A-CATs [6]: A-CATs employs an actor-critic approach to train TSC agents, where the output phase duration is adjusted within a range of 10 to 40 seconds. This method provides continuous learning for phase duration optimization based on traffic conditions.

- 3DQN-TSCC [7]: 3DQN-TSCC applies DQN to adjust phase durations in small increments, focusing on stabilizing the signal light transitions. In this method, the phase duration is modified by a fixed set of values $\{-5, 0, 5\}$ s.

- CCDA [10]: CCDA introduces a centralized critic and decentralized actor framework, ensuring stability in phase duration changes. The method adjusts all phase durations in smaller steps $\{-6, -3, 0, 3, 6\}$ s and decisions are made every 10 s to ensure stability.

**VLM-Based Method.** We also consider a VLM-based approach, Vanilla-VLM, which directly utilizes a VLM for scene understanding and generates a textual description of the traffic situation, which is then used by an LLM to make decisions without the involvement of RL policies in regular scenarios.

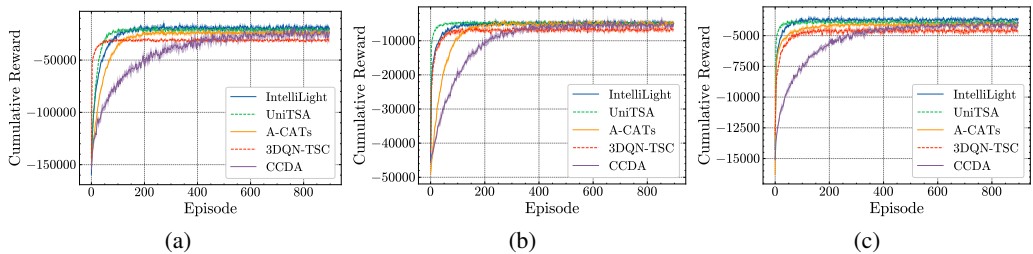

Figure 12: Training reward curves of five RL-based TSC methods across three real-world intersections: (a) Songdo (South Korea), (b) Yau Ma Tei (Hong Kong), and (c) Massy (France).

# C    Additional Experiments Results

## C.1    Additional Performance Analysis of RL Methods for TSC

In this section, we describe five RL-based TSC methods: IntelliLight, UniTSA, A-CATs, CCDA, and 3DQN-TSC. Figure 12 shows the reward curves for each method in three scenarios, where the x-axis represents training episodes and the y-axis represents cumulative rewards. Among the methods, IntelliLight and UniTSA achieve the highest cumulative rewards and exhibit rapid convergence. Their superior performance stems from their design: both adopt direct phase-switching actions, allowing for timely and responsive adjustments to dynamic traffic conditions.

A-CATs and CCDA exhibit competitive, though slightly inferior performance. A-CATs decomposes the multi-phase scheduling task into sequential single-phase adjustments, which increase the learning difficulty but eventually result in near-optimal control once convergence is reached. CCDA extends this idea by enabling simultaneous updates of all phases with a larger action space, leading to slower convergence but similar final performance.

Finally, 3DQN-TSC performs the worst across all scenarios. Its restrictive action design—limited to modifying a single phase per cycle—constrains its ability to optimize phase allocations holistically. As a result, it accumulates fewer rewards and struggles to match the performance of other methods.

Table 5: Performance comparison of different VLMs for scene understanding. The top two results are marked with $*$ (best), $\dagger$ (second).

| Scene | Metrics | VLMLight | Qwen2.5-VL-7B | LLava-7B | LLava-13B | GPT-4o |
|---|---|---|---|---|---|---|
| Songdo | ATT ↓ | $87.14^*$ | 92.68 | 95.25 | 91.96 | $87.46^\dagger$ |
| Yau Ma Tei | | $39.80^*$ | 41.79 | 48.54 | 41.27 | $41.12^\dagger$ |
| Massy | | $60.84^*$ | 65.62 | 70.54 | 66.44 | $63.72^\dagger$ |
| Songdo | AETT ↓ | $49.88^\dagger$ | 76.13 | 80.45 | 60.86 | $48.06^*$ |
| Yau Ma Tei | | $13.50^\dagger$ | 30.34 | 27.55 | 19.71 | $13.04^*$ |
| Massy | | $45.40^\dagger$ | 62.63 | 65.97 | 55.78 | $45.17^*$ |

Table 6: Performance comparison of different LLMs for mode selection.

| Scene | Metrics | VLMLight | Qwen2.5-7B | Qwen2.5-32B | Llama3-70B | GPT-4o |
|---|---|---|---|---|---|---|
| Songdo | ATT ↓ | $87.14^\dagger$ | 87.18 | $86.28^*$ | 89.53 | 88.57 |
| Yau Ma Tei | | $39.80^\dagger$ | 41.01 | 41.14 | $38.89^*$ | 40.19 |
| Massy | | $60.84^\dagger$ | 62.46 | 62.73 | 62.67 | $60.79^*$ |
| Songdo | AETT ↓ | 49.88 | $49.81^\dagger$ | 50.67 | 51.25 | $49.70^*$ |
| Yau Ma Tei | | 13.50 | 13.36 | 13.91 | $13.19^\dagger$ | $12.96^*$ |
| Massy | | $45.40^*$ | 46.81 | 48.61 | $45.44^\dagger$ | 46.76 |

Table 7: Performance comparison of different LLMs on reasoning policy.

| Scene | Metrics | VLMLight | Qwen2.5-7B | Qwen2.5-32B | Llama3-70B | GPT-4o |
|---|---|---|---|---|---|---|
| Songdo | ATT ↓ | $87.14^*$ | 90.59 | 88.59 | $87.40^\dagger$ | 87.99 |
| Yau Ma Tei | | 39.80 | $38.50^\dagger$ | 40.57 | $38.23^*$ | 39.93 |
| Massy | | $60.84^\dagger$ | 63.71 | 62.56 | $59.11^*$ | 61.11 |
| Songdo | AETT ↓ | 49.88 | 51.85 | 49.71 | $49.45^\dagger$ | $49.37^*$ |
| Yau Ma Tei | | $13.50^\dagger$ | 13.34 | 13.97 | 13.76 | $13.14^*$ |
| Massy | | $45.40^\dagger$ | 53.68 | 46.68 | 50.54 | $44.61^*$ |

## C.2 Additional Ablation Study on different LLMs

In this section, we analyze the impact of different LLM models on the performance of VLMLight across various modules, including scene understanding, mode selection, and reasoning policy. We evaluated multiple models, including Qwen2.5-VL-7B [29], LLaVA-7B, LLaVA-13B [44], and GPT-4o [20] for the scene understanding agent ($\text{Agent}_{\text{Scene}}$), and Qwen2.5-7B, Qwen2.5-32B [30], Llama3.1-70B [45], and GPT-4o [20] for the $\text{Agent}_{\text{ModeSelector}}$ and reasoning policy agents. The results are presented in Tables 5, 6, and 7, showcasing the effects of model selection on the respective modules.

The results indicate that the scene understanding module ($\text{Agent}_{\text{Scene}}$) is the most sensitive to model changes. As shown in Table 5, the performance of the model significantly affects both the Average Travel Time (ATT) and Average Emergency Travel Time (AETT), especially in identifying special vehicles. For instance, switching from Qwen2.5-VL-32B to Qwen2.5-VL-7B results in notable increases in the waiting time and travel time of special vehicles, likely due to missed recognition of emergency vehicles. Moreover, the performance drop in model accuracy also leads to longer travel times for regular vehicles, as normal vehicles may be mistakenly treated as special vehicles, causing unnecessary green lights to be given. This highlights the importance of a reliable and accurate scene understanding for the overall system performance, as inaccurate scene descriptions can propagate errors in the subsequent decision-making stages.

In contrast, the $\text{Agent}_{\text{ModeSelector}}$ and reasoning policy agents, which involve simpler textual processing tasks, show more resilience to model changes. As indicated in Table 6 and Table 7, even smaller models like Qwen2.5-7B maintain similar performance to larger models, with only marginal differences in ATT for regular vehicles. However, special vehicles may still experience slight delays due to inaccurate lane-to-phase mappings in the reasoning process. Overall, the most critical takeaway is that the scene understanding module has the greatest impact on VLMLight's performance, particularly in ensuring timely prioritization of special vehicles. Thus, using a high-performance model for scene understanding is essential for maintaining the system's ability to handle complex, safety-critical scenarios effectively.

## C.3 Additional Ablation Study on Inference Time

In this section, we analyze the inference time of VLMLight across three distinct environments. The results, shown in Table 8, demonstrate that VLMLight achieves inference times well below 13 s in all three environments, falling within an acceptable range for real-world deployment. This is particularly notable considering the minimum green light duration of 10 s and the additional 3 s for yellow lights.

As illustrated in Table 8, the majority of the inference time is spent on scene understanding, while mode selection and deliberative reasoning stages require considerably less time. Overall, the results indicate that VLMLight is suitable for deployment in practical settings, with its architecture optimized for both speed and safety.

Table 8: Inference time for each stage of VLMLight across three environments.

| Stage | Songdo | Yau Ma Tei | Massy |
|---|---|---|---|
| $\text{Agent}_{\text{Scene}}$ | 5.12 | 5.15 | 4.79 |
| $\text{Agent}_{\text{ModeSelector}}$ | 0.75 | 0.95 | 0.77 |
| $\text{Agent}_{\text{Phase}}$ | 3.87 | 1.95 | 2.24 |
| $\text{Agent}_{\text{Plan}}$ | 1.23 | 0.86 | 1.21 |
| $\text{Agent}_{\text{Check}}$ | 0.51 | 0.45 | 0.34 |
| Total | **11.48** | **9.36** | **9.35** |

# D  Case Study

To showcase **VLMLight** in action, we present three representative case studies. Each example covers a complete TSC cycle from time step $T$ to $T+1$, demonstrating how different agents collaborate under both routine and safety-critical scenarios. For each case, we describe the visual inputs, the decision

made by each agent, and the resulting traffic outcome. This offers insight into how VLMLight dynamically selects between the fast RL branch and the deliberative LLM branch as circumstances demand.

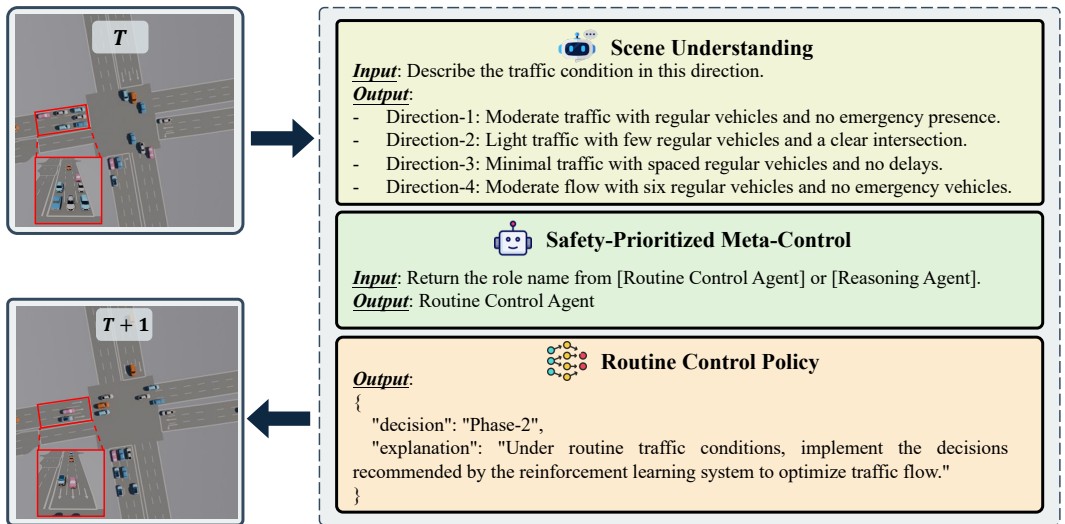

Figure 13: Routine Control in Yau Ma Tei.

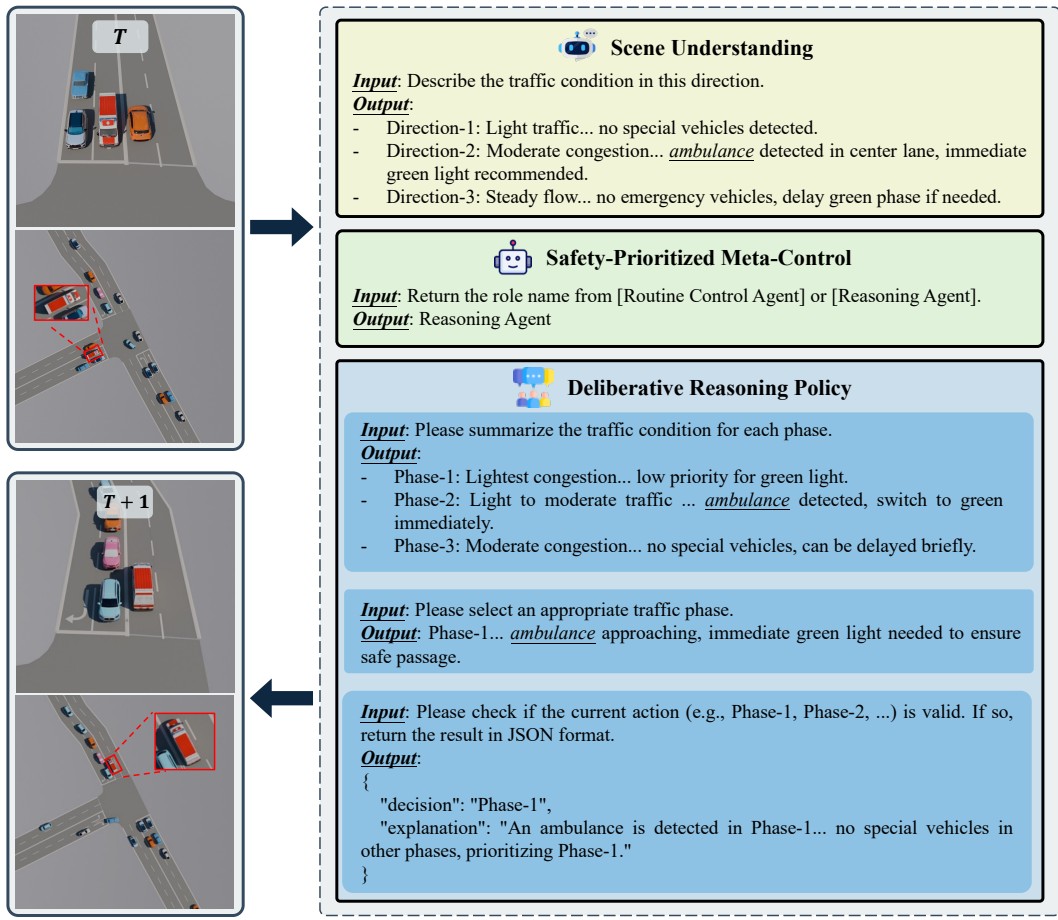

Figure 14: Deliberative Reasoning policy for complex traffic in Massy.

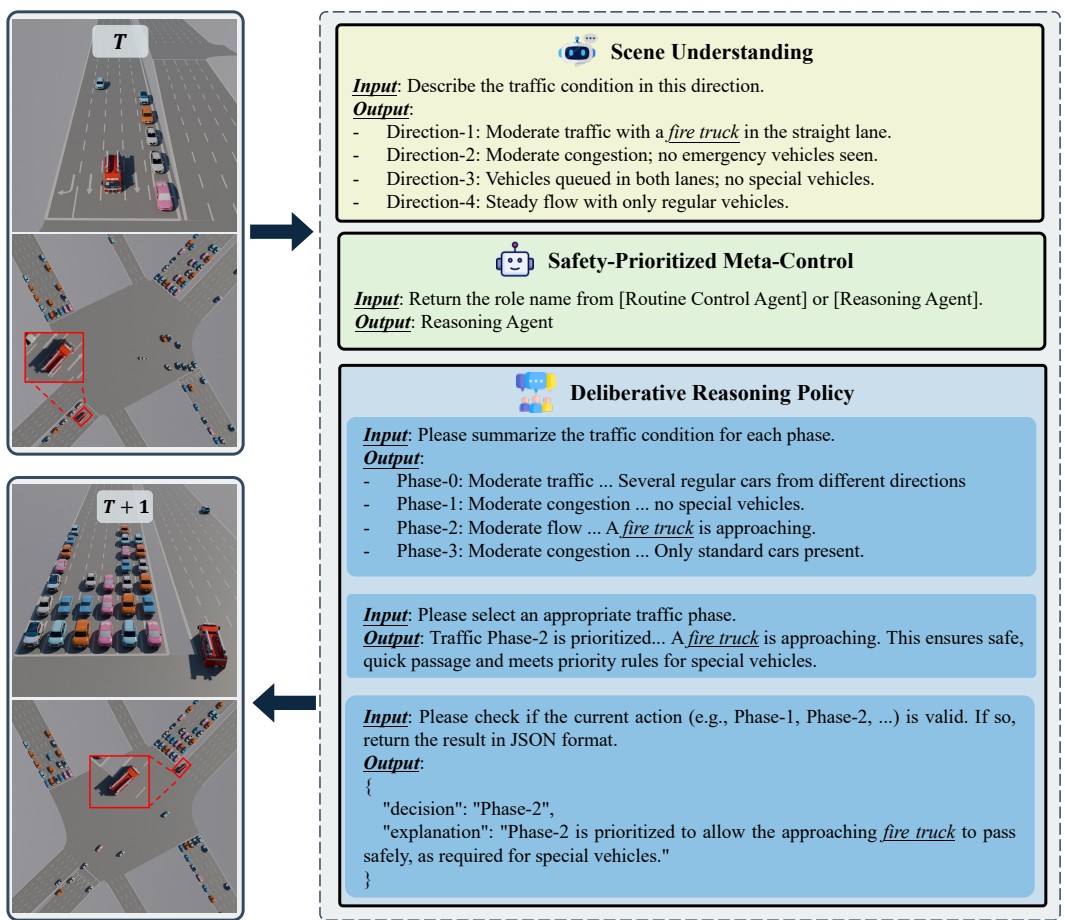

Figure 15: Deliberative Reasoning policy for complex traffic in Songdo.

### D.1 Example 1: Routine Control in Yau Ma Tei

Figure 13 presents a routine scenario at the Yau Ma Tei intersection. The $\text{Agent}_{\text{Scene}}$ first transforms multi-view traffic images into structured language descriptions. Finding no anomalies or priority vehicles, the $\text{Agent}_{\text{ModeSelector}}$ routes the control to the RL branch. Based on the traffic density, the $\text{Agent}_{\text{RL}}$ selects Phase-2 (westbound) for the green signal. The transition from $T$ to $T+1$ confirms that the westbound queue clears once Phase 2 is activated.

### D.2 Example 2: Complex Scenario in Massy

Figure 14 showcases a special case at the Massy intersection, where an ambulance is detected on the west approach. The $\text{Agent}_{\text{Scene}}$ detects the emergency vehicle from the image inputs and generates descriptive observations. Recognizing a priority event, the $\text{Agent}_{\text{ModeSelector}}$ subsequently triggers the Deliberative Reasoning branch. The $\text{Agent}_{\text{Phase}}$ agent maps the scene to candidate signal phases, $\text{Agent}_{\text{Plan}}$ recommends Phase-1 for a green signal, and $\text{Agent}_{\text{Check}}$ verifies compliance with emergency-priority rules. By the time $T+1$, the ambulance has cleared the intersection through the northbound turn.

### D.3 Example 3: Complex Scenario in Songdo

Figure 15 presents a complex case at the Songdo intersection. Similar to the previous Massy case, the $\text{Agent}_{\text{Scene}}$ identifies key cues (a fire truck approaching), and $\text{Agent}_{\text{ModeSelector}}$ activates the Deliberative Reasoning branch. The sequence of $\text{Agent}_{\text{Phase}}$, $\text{Agent}_{\text{Plan}}$, and $\text{Agent}_{\text{Check}}$ ensures a

compliant and safe control action. By the time $T + 1$, the fire truck moves through the intersection without interruption.

# E    Disscussion

VLMLight introduces a novel vision-language framework for TSC, combining real-time visual reasoning with safety and efficiency improvements. As the first open-source vision-based simulator in the TSC domain, VLMLight is compatible with RL-based TSC algorithms, offering a valuable resource for future research on perception-driven traffic systems. This framework enables enhanced scene understanding through multi-view visual perception and structured reasoning, ensuring both fast decision-making for routine traffic and reliable handling of critical scenarios like emergency vehicles.

However, VLMLight has several limitations. Firstly, the current simulator lacks diverse weather and lighting conditions, limiting its evaluation of visual robustness in challenging environments. Secondly, the absence of pedestrians, bicycles, and other real-world elements makes the simulated environment less realistic; incorporating more diverse models is necessary for future iterations. Third, all experiments have been conducted on single intersections, and extending the framework to multi-intersection scenarios requires further research on scalability and coordination in broader traffic networks. Lastly, VLMLight's performance is closely tied to VLM capabilities, with optimal results requiring models with numerous parameters. Future work will focus on fine-tuning smaller models to improve both traffic scene recognition accuracy and inference speed.

