# OpenReview forum: "VLMLight: Safety-Critical Traffic Signal Control via Vision-Language Meta-Control and Dual-Branch Reasoning Architecture"
_NeurIPS.cc/2025/Conference — NeurIPS 2025 poster_

### Official Review · Reviewer_Srph · 2025-06-15

**Clarity:** 3
**Significance:** 2
**Originality:** 2
**Rating:** 3
**Confidence:** 3

**Summary:**

This paper introduces VLMLight, a Traffic Signal Control (TSC) framework that integrates vision-language meta-control with dual-branch LLM-based reasoning. In particular, it first uses  VLM to process multi-view images to detect the existence of a special vehicle to route to either a fast RL policy for routine traffic or a structured LLM-based reasoning branch for critical cases. The authors evaluate VLMLight against rule-based and RL-based TSC on data collected from three real-world intersections and show the reduction of waiting times for emergency vehicles by up to 65% over RL-only systems with a performance degradation in standard conditions for less than 1%.

**Questions:**

-can the method be adjusted to not degrade the regular traffic waiting time while still decreasing the special vehicles waiting time to some extent?

-how do the use of VLMs impact the results?

**Ethical Concerns:**

["NO or VERY MINOR ethics concerns only"]

**Final Justification:**

I am not fully convinced by the author's response and still have some concerns regarding practicalness of the current approach because of the 1% compromise on regular traffic (which takes the most majority of the traffic) and the lack of evaluation in multi-intersection networks.

**Limitations:**

yes.

**Quality:**

2

**Strengths And Weaknesses:**

Pros

-well-written and clear

-effectively reduce the waiting of special vehicles

Cons

-the proposed method (using VLM to select between RL-based TSC and LLM-based TSC) is practical regarding the evaluated metrics (waiting time for regular vehicle/special vehicle) but relatively incremental.

-the practicalness/impact of the proposed method can also be potentially limited as in practice 1% decrease of waiting time of regular vehicles can be significant due to their absolute majority.

---

> ### Author Rebuttal · Authors · 2025-07-31
>
> We sincerely thank the reviewer for their constructive feedback and address each of the concerns and questions below.
>
> > [W1] On the concern that the method is "relatively incremental."
>
> We understand your concern. However, we would like to emphasize that VLMLight introduces two key innovations:
> - The first **image-based traffic simulator for TSC** and support multi-view intersection perception, enabling policies to reason about vehicle type, motion, and spatial density.
> - A vision-language meta-control framework that generalizes beyond emergency vehicles. While emergency vehicle priority is used as a primary case study, the same mechanism can handle other safety-critical events such as pedestrian safety, public transport lanes.
>
> Moreover, the impact is substantial: emergency vehicle waiting times dropped from 52s to just 2s (Table 1), while maintaining real-time performance. This goes well beyond what rule-based or RL-only approaches have demonstrated.
>
> > [W2] On the impact of a ~1% increase in regular vehicle waiting time.
>
> We acknowledge that VLMLight introduces a small ~1% increase in waiting time for regular vehicles when emergency vehicles are present. This is an expected trade-off, as the system must temporarily reprioritize traffic flow to ensure that emergency vehicles can pass safely and quickly [1].
>
> Importantly, VLMLight maintains RL-only performance for regular traffic when no emergency vehicles are present. In addition, prior studies [2,3] focused only on emergency vehicle delay without reporting the impact on regular vehicles. In contrast, our work evaluates both sides, offering a transparent view of this trade-off.
>
> We also note that the dual-branch design is modular, the fast RL branch can be replaced with an improved policy in future work, potentially further reducing this minor impact.
>
> > [Q1] Can the method be adjusted to avoid degrading regular traffic?
>
> Our current system already ensures that in the absence of special vehicles, VLMLight relies solely on the RL branch, achieving state-of-the-art regular traffic performance. When emergency vehicles are detected, a slight compromise (1%) is made for regular vehicles to enable rapid clearance of priority vehicles. This trade-off is not only inevitable but consistent with safety priorities in real-world traffic management.
>
> That said, our architecture is designed for easy upgrade: future RL policies, heuristic scheduling, or hybrid optimization methods could be plugged into the "fast branch," allowing improvements in regular vehicle flow while preserving the core safety benefits.
>
> > [Q2] How do VLMs impact the results?
>
> VLMs are central to VLMLight's success. They provide multi-view semantic understanding of intersections, detecting special events like the arrival of an emergency vehicle.
>
> As shown in Appendix Table 3, larger VLMs significantly outperform smaller ones in scene interpretation accuracy, directly improving decision quality. This highlights that VLM-powered perception is not merely a detector replacement; it enables structured, interpretable reasoning over complex traffic scenes that conventional computer vision pipelines cannot achieve.
>
> Reference
>
> [1] Nguyen, Dang Viet Anh, et al. "Robustness of Reinforcement Learning-Based Traffic Signal Control under Incidents: A Comparative Study." arXiv preprint arXiv:2506.13836 (2025).
>
> [2] Zhong, Li, and Yixiang Chen. "A novel real-time traffic signal control strategy for emergency vehicles." IEEE Access 10 (2022): 19481-19492.
>
> [3] Cao, Miaomiao, Victor OK Li, and Qiqi Shuai. "A gain with no pain: Exploring intelligent traffic signal control for emergency vehicles." IEEE Transactions on Intelligent Transportation Systems 23.10 (2022): 17899-17909.

---

> > ### Comment · Reviewer_Srph · 2025-08-05
> > **Thank you for the response!**
> >
> > Thanks for the response! While I appreciate the author's transparency on the trade-off, I still have some concerns regarding the 1% compromise on regular traffic as emergency traffic accounts for even less than 1% of all traffic -> the average waiting time of all vehicles increases. In practice, when the emergency vehicles use their siren, it already has similar effect to some extent so it is hard to justify if this 1% compromise worths it.
> > Additionally, the point regarding scalability to multi-intersection networks raised by Reviewer 6Mxr also concerned me as this further reduce the practical impact of the current work.
> > Because of these reasons, I hold a more borderline position regarding the current paper.

---

> > > ### Author Response · Authors · 2025-08-05
> > >
> > > We appreciate the opportunity to engage in this conversation and thank the reviewer for the thoughtful feedback. Below, we respond to the two concerns raised, separately.
> > >
> > > > About the 1% compromise on regular traffic for emergency vehicle prioritization
> > >
> > > Thank you for raising this important point. Although emergency vehicles account for less than 1 percent of total traffic, their functions, such as transporting critically ill patients or responding to fires, **carry significant societal importance**. We believe that a coordinated system-level response is essential, especially in complex urban settings where sirens alone may not ensure effective traffic clearance. For example, in cases of heavy congestion, an emergency vehicle may become stuck behind long queues and be unable to reach the intersection even when using lights and sirens. In contrast, our system can proactively clear relevant lanes through signal control before the vehicle reaches the intersection, ensuring a faster and safer passage.
> > >
> > > In our system, the 1% overall compromise leads to an average delay increase of only ~0.34 seconds. This small cost yields significant benefits for emergency response time, which we argue is a worthwhile and necessary trade-off in safety-critical settings.
> > >
> > > Moreover, VLMLight is designed to handle not only emergency vehicles but also a broader class of rare but critical events. Our simulator supports diverse and realistic scenarios such as fallen obstacles or abnormal pedestrian behavior. The reasoning branch in VLMLight enables adaptability in such cases, which traditional RL-based methods struggle to handle due to fixed training objectives and limited exposure to long-tail events.
> > >
> > > > About scalability to multi-intersection networks
> > >
> > > We agree that scalability is essential for real-world deployment. **VLMLight is modular by design and supports flexible integration into larger traffic control systems.** For example, the fast RL branch can be replaced with decisions generated by multi-agent RL frameworks to enable coordinated control across multiple intersections.
> > >
> > > The vision-language reasoning component is also adaptable. Depending on the deployment environment, different model sizes can be selected based on the available hardware and the latency requirements of specific intersections. This flexibility allows VLMLight to maintain real-time performance while ensuring that safety-critical reasoning is available where and when it is most needed.

---

### Official Review · Reviewer_6Mxr · 2025-07-02

**Clarity:** 3
**Significance:** 3
**Originality:** 3
**Rating:** 3
**Confidence:** 4

**Summary:**

VLMLight introduces a vision-language framework for traffic signal control that addresses part of the limitations of traditional rule-based and RL-only methods. The framework leverages a custom traffic simulator that supports multi-view visual inputs for real-time perception of vehicle types, spatial density, and dynamic motion patterns. LLM and VLM agents act as a meta-controller to dynamically switch between a fast RL policy and a deliberative reasoning branch for safety-critical scenarios. Experimental results demonstrate a feasible reduction in emergency vehicle waiting times while maintaining comparable efficiency in standard conditions.

**Questions:**

- **Justification for Latency**: The authors state that the system’s 11.5-second end-to-end runtime falls within an acceptable 13-second urban signal buffer. However, no citation or empirical evidence is provided to support this threshold. How was the 13-second buffer determined? Please provide a reference or data-driven justification to substantiate this claim.

**Ethical Concerns:**

["NO or VERY MINOR ethics concerns only"]

**Final Justification:**

concerning the effectiveness-efficiency tradeoff of current model, I maintain my score

**Limitations:**

See weakness and questions

**Quality:**

3

**Strengths And Weaknesses:**

## Strengths

- **Safety-Critical Adaptability**: The dual-branch architecture ensures fast execution for routine traffic while activating dedicated decision-making for emergencies. This hybrid approach reduces the waiting time for emergency vehicles. The author successfully proposed an effective approach to addressing this gap in prior TSC systems.
- **Open-Source Simulator for Vision-Based TSC**: The release of a vision-based simulator with dynamic multi-view rendering enables future research on perception-driven traffic control, fostering community innovation. While there are alternatives around [1], the authors may over claim their contribution as the **first** vision-based simulator.
- **Innovative Integration of Vision and Language Models**: VLMLight is the first to combine visual scene understanding with LLM reasoning in TSC, enabling richer semantic analysis of traffic dynamics compared to traditional RL-based state representations.

## Weakness

1. **Scalability to multi-intersection networks**: The experiments are confined to single intersections. While VLMLight reduces emergency vehicle waiting time locally, its performance in coordinated multi-intersection scenarios—critical for real-world emergency response efficiency—remains untested. The framework’s reliance on high-fidelity visual inputs and LLM deliberation may introduce latency bottlenecks when scaled to city-wide deployments.
2. **Justification for vision-language overhead**:
   - **Redundancy in perception**: The paper does not include ablation studies comparing vision-language models (e.g., Qwen-VL, LLaVA) with simpler object detection approaches (e.g., YOLO) for emergency vehicle recognition. While VLMs offer richer semantic understanding, their computational overhead—approximately 5.12 seconds per inference (Table 8)—is significantly higher than detection-only models. It remains unclear whether such complexity is necessary, given that traditional object detectors may be sufficient for accurate emergency vehicle identification.
   - **Cost-effectiveness**: The paper lacks discussion on the deployment costs associated with maintaining dual VLM/LLM stacks, potentially requiring high-end GPUs (e.g., A100) per intersection. Rule-based or reinforcement learning methods with emergency prioritization heuristics could offer comparable performance with substantially lower resource requirements. Additional experiments and rationale are needed to justify the proposed approach's cost and design complexity.

[1] Jin, Zhijing, Tristan Swedish, and Ramesh Raskar. "3d traffic simulation for autonomous vehicles in unity and python." *arXiv preprint arXiv:1810.12552* (2018)

---

> ### Author Rebuttal · Authors · 2025-07-31
>
> We sincerely thank the reviewer for the thoughtful and constructive comments. Below, we respond point by point.
>
> > [W1] Scalability to Multi-Intersection Networks
>
> We fully agree that multi-intersection coordination is an essential challenge in real-world TSC. Our decision to begin with single-intersection settings was driven by three reasons:
> - **Foundational Importance**, as highlighted by prior surveys [1,2], single-intersection control is the fundamental unit of TSC research and remains a crucial benchmark before scaling to larger networks.
> - **Modular Design for Generalization**, VLMLight is intentionally built as a modular framework: the vision-language meta-controller and dual-branch reasoning can be plugged into any multi-intersection coordination algorithm (e.g., Max-Pressure or MARL approaches). In fact, our experiments already include three distinct intersection geometries, demonstrating that the framework adapts to varied layouts without retraining.
> - We also acknowledge the reviewer's latency concern when scaling to larger networks. Recent studies [3,4,5] show that LLMs can now operate efficiently on edge devices, and our validation with the lightweight Gemma3 [6] model further confirms this. VLMLight maintains real-time performance with minimal degradation even under resource constraints, as shown below:
>
> **Performance Comparison (seconds)**
> |           |      | Songdo | Yau Ma Tei | Massy |
> |:---------:|------|:------:|:----------:|-------|
> | Gemma3-4b | ATT  |  89.81 |    41.38   | 64.79 |
> |           | AETT |  47.54 |    14.57   | 47.01 |
> |  VLMLight | ATT  |  87.14 |    39.80   | 60.84 |
> |           | AETT |  49.88 |    13.50   | 45.41 |
>
> > [W2] Justification for Vision-Language Overhead
>
> We sincerely thank the reviewer for raising this important point. To address the concern about vision-language overhead, we provide a two-part justification: first explaining why a VLM is necessary, then clarifying how VLMLight minimizes reasoning and deployment costs.
>
> **(a) Why not rely on simpler object detection approaches (e.g., YOLO)?**
> Our objective extends beyond merely detecting emergency vehicles, we aim for scene-level reasoning to support safety-critical decisions. Traditional detectors like YOLO require lane-specific annotations and manual direction labeling for every intersection [7], making them brittle when adapting to new layouts. In contrast, VLMLight's vision-language module:
> - Adapts seamlessly to unseen intersection geometries without re-labeling;
> - Understands contextual anomalies, such as a fallen bicycle blocking a lane or an ambulance stopped in a non-standard position, reasoning that goes beyond binary "is this an emergency vehicle?" detection.
>
> **(b) Deployment cost and dual-stack complexity.**
> We fully agree that efficiency is crucial for real-world deployment. VLMLight addresses this through its dual-branch architecture:
> - The fast RL branch handles the vast majority of routine traffic decisions, ensuring real-time performance.
> - The structured reasoning branch is only activated in rare, safety-critical scenarios (e.g., emergency vehicle prioritization), minimizing computational overhead.
>
> Moreover, VLMLight's modular design allows the VLM/LLM components to be swapped for lighter alternatives (e.g., Gemma3 or other models) as hardware constraints demand. Our released image-based simulator further empowers future researchers to experiment with different model sizes, balancing cost vs. accuracy without redesigning the entire framework.
>
> > [Q1] Latency Justification
>
> We thank the reviewer for highlighting this important issue. To clarify why VLMLight's latency is acceptable for real-world deployment, we address it in three parts:
>
> **(a) Safety standards create a natural timing buffer.**
> Pedestrian safety regulations require a minimum green interval to ensure safe crossing. Using the standard calculation formula from [8,9] (assuming a walking speed of 1.2m/s), the required pedestrian green times for our three test intersections are 25s, 23s, and 17s (based on crosswalk widths of 30m, 25m, and 17m). In addition, government guidelines (e.g., South Australia [10], Toronto [11]) specify a minimum 9s green phase plus 3s yellow, effectively providing at least a 12s reasoning buffer before any signal phase change.
>
> **(b) Latency mainly applies to rare, complex scenarios.**
> VLMLight's 11.5 s latency occurs only in the infrequent, safety-critical cases that require multi-agent deliberation. In the majority of traffic flow situations, the fast RL + VLM branch is used, which operates significantly faster.
>
> **(c) Lightweight model substitution further reduces latency.**
> VLMLight is modular, meaning the perception and reasoning agents can be replaced with lighter models. For example, when we substituted the VLM/LLM components with Gemma3-4B, inference latency shows in the following table. These results confirm that even in the most demanding scenarios, VLMLight stays well within the available safety margin, while in regular cases inference time is typically under 2s with lightweight models.
>
> **Inference Latency (seconds)**
> |           |         | Songdo | Yau Ma Tei | Massy |
> |:---------:|---------|:------:|:----------:|-------|
> | Gemma3-4b | Normal  |  1.61  |    1.49    | 1.45  |
> |           | Safety-critical|  2.38  |    2.29    | 2.21  |
> |  VLMLight | Normal  |  5.87  |    6.10    | 5.56  |
> |           | Safety-critical|  11.48 |    9.36    | 9.55  |
>
> This means VLMLight remains well within safety margins, and we have added these references and calculations to the manuscript.
>
> Reference
>
> [1] Wei, Hua, et al. "A survey on traffic signal control methods." arXiv preprint arXiv:1904.08117 (2019).
>
> [2] Zhao, Haiyan, et al. "A survey on deep reinforcement learning approaches for traffic signal control." Engineering Applications of Artificial Intelligence 133 (2024): 108100.
>
> [3] Xu, Jiajun, et al. "On-device language models: A comprehensive review." arXiv preprint arXiv:2409.00088 (2024).
>
> [4] Friha, Othmane, et al. "Llm-based edge intelligence: A comprehensive survey on architectures, applications, security and trustworthiness." IEEE Open Journal of the Communications Society (2024).
>
> [5] Yao, Yuan, et al. "Efficient GPT-4V level multimodal large language model for deployment on edge devices." Nature Communications 16.1 (2025): 5509.
>
> [6] Team, Gemma, et al. "Gemma 3 technical report." arXiv preprint arXiv:2503.19786 (2025).
>
> [7] Lin, Jia-Ping, and Min-Te Sun. "A YOLO-based traffic counting system." 2018 Conference on Technologies and Applications of Artificial Intelligence (TAAI). IEEE, 2018.
>
> [8] Koonce, Peter. Traffic signal timing manual. No. FHWA-HOP-08-024. United States. Federal Highway Administration, 2008.
>
> [9] Iryo-Asano, Miho, Wael KM Alhajyaseen, and Hideki Nakamura. "Analysis and modeling of pedestrian crossing behavior during the pedestrian flashing green interval." IEEE Transactions on Intelligent Transportation Systems 16.2 (2014): 958-969.
>
> [10] Government of South Australia, Traffic Signal Standard: Signal Timings - TS001
>
> [11] City of Toronto, Standard Operating Practice: Pedestrian Timing at Signalised Intersections. Transportation Services

---

> > ### Comment · Reviewer_6Mxr · 2025-08-04
> > **ack**
> >
> > Thanks for the additional results on latency and performance against general vlm. It seems the marginal performance gain is a tradeoff of additional inference latency?

---

> > > ### Author Response · Authors · 2025-08-05
> > >
> > > Thank you for your insightful follow-up. We appreciate the opportunity to further clarify our design decisions and experimental findings. While larger models generally offer stronger scene understanding, **VLMLight is explicitly designed to balance performance and latency through flexible deployment**. The architecture of the proposed framework allows dynamic choosing of the model size based on intersection complexity, real-time constraints, and hardware availability.
> > >
> > > We also emphasize that VLMLight's architecture enables selective and situational-aware reasoning, rather than always incurring additional inference latency.
> > >
> > > Finally, the image-based traffic simulator we developed allows for the creation of rich, realistic urban scenarios where vision-language reasoning becomes crucial, extending the practical applicability of TSC research.

---

### Official Review · Reviewer_biB6 · 2025-07-02

**Clarity:** 3
**Significance:** 2
**Originality:** 2
**Rating:** 4
**Confidence:** 4

**Summary:**

The paper presents a vision-language model-based approach for traffic signal control. A novel custom-built traffic simulator is first proposed, and multiple LLM agents are introduced and used as controllers to reduce the waiting time of emergency vehicles.

**Questions:**

1. Maybe I understand the results wrong - I am wondering whether the authors have reported the real-time performance of the proposed approach. Given that the approach involves vision-language model, and multiple LLM agents, are all LLMs deployed in local and the inference time can be ignored? Or should the approach wait for the responses of the LLMs? If so, the overall computation time of the approach cannot meet the real-time requirement in real-time traffic control application.

2. Is it possible that we directly use the embeddings along the process instead of using explicit texts, as texts sometimes can be misleading and inappropriate representations can affect the model performance?

3. How can we guarantee that the LLM agents (controllers) can always give safe instructions in order to prioritize the emergency vehicles? I understand that if the actions are not safe, we will select an alternative, but do the authors conduct any operation to guarantee that there will be an alternative that is feasible?

4. The overall framework, while unseen before in my opinion, is still a bit trivial. Basically we just let the VLM tell us what happen and LLM agents respond to prioritize the emergency vehicles. Also, the experimental results only support that the approach is better when there is emergency vehicle in the intersections. Somehow this feels limited and cannot be generalized to other scenarios. Please correct me if I am wrong.

**Ethical Concerns:**

["NO or VERY MINOR ethics concerns only"]

**Final Justification:**

I have read the rebuttal and I think the responses have addressed some of my concerns. I have raised my scores accordingly.

**Limitations:**

yes

**Paper Formatting Concerns:**

N.A.

**Quality:**

3

**Strengths And Weaknesses:**

Strength:
1. A new traffic simulator
2. A meaningful and practical problem setting involving emergency vehicles.
Weakness:
1. The reliance of scene description and phase description in terms of text seems a bit unnecessary, which may even introduce noise since text representations can be misleading sometimes.
2. While important, the improvement only for emergency vehicles is still limited.

---

> ### Author Rebuttal · Authors · 2025-07-31
>
> We thank the reviewer for the thoughtful and insightful comments. We respond to each of the points as follows:
>
> > [W1]. Use of textual scene/phase descriptions may introduce noise
>
> We appreciate the reviewer's point and agree that text representations require careful design. **Our intent for using textual descriptions is to ensure interpretability, traceability, and trustworthiness-critical factors for deployment in real-world traffic systems.** In real-world traffic signal control, these aspects are indispensable. Textual descriptions allow every decision made by the system to be easily reviewed, audited, and justified by traffic authorities, capabilities that are difficult to achieve with latent embeddings alone. Prior work [1] has also emphasized the importance of interpretability in TSC, reinforcing our design choice.
>
> Crucially, all final actions are subject to a rule-based compliance check that strictly enforces traffic regulations and safety constraints. This mechanism ensures that even if noise were introduced at the textual stage, only safe and legally valid traffic signal actions can be executed, thereby eliminating the risk of unsafe outcomes from LLMs.
>
> > [W2]. Improvement only on emergency vehicles feels limited
>
> We agree that the current experiments focus on emergency vehicle scenarios, but we respectfully emphasize that emergency vehicle handling is a core safety-critical issue in traffic signal control research, as demonstrated by prior work such as EMVLight [2]. Our approach enables real-time emergency vehicle prioritization **directly from intersection camera feeds**, without relying solely on pre-collected statistical data.
>
> Additionally, we open-sourced the **image-based traffic signal control simulator**, shifting TSC research from purely statistical inputs to realistic vision-based scenarios, which we believe will benefit a broader range of future studies and applications.
>
> > [Q1]. Real-time feasibility of multi-agent LLM reasoning
>
> We agree that real-time performance is crucial. VLMLight addresses this by design:
> 1. Minimum green time inherently provides reasoning time. In TSC, **pedestrian safety** necessitates a minimum green light interval. Based on standard calculation formulas cited in [3,4], the minimum pedestrian green times for our three experimental intersections are 25s, 23s, and 17s (corresponding to crosswalk widths of 30m with a median island, 25m, and 17m respectively). As shown in Table 2, VLMLight's reasoning latency, around 10s (even when both VLM and LLM are engaged) remains well below these green time thresholds. Moreover, government guidelines (e.g., Government of South Australia [5], Toronto [6]) stipulate minimum pedestrian green times of 9s for all users (including those using mobility aids). Adding the standard 3s yellow phase, there is a guaranteed 12s buffer for reasoning, which comfortably covers VLMLight's inference time.
> 2. Multi-agent reasoning is only triggered in critical scenarios. The RL branch handles routine traffic control, while the multi-agent LLM reasoning branch is activated only in safety-critical cases (e.g., emergency vehicles, obstructions). This design means that most cycles involve only the VLM scene perception and mode selector decision, avoiding multi-round LLM reasoning in the majority of situations.
> 3. To further confirm real-world deployability, we evaluated VLMLight using Gemma3-4B [7], a lightweight vision-language model optimized for edge devices. As shown below, Gemma3-4B's inference time is only ~1.5s in normal cases and <3s even in safety-critical scenarios, easily within operational limits.
>
> **Inference Latency (seconds)**
> |           |         | Songdo | Yau Ma Tei | Massy |
> |:---------:|---------|:------:|:----------:|-------|
> | Gemma3-4b | Normal  |  1.61  |    1.49    | 1.45  |
> |           | Safety-critical |  2.38  |    2.29    | 2.21  |
> |  VLMLight | Normal  |  5.87  |    6.10    | 5.56  |
> |           | Safety-critical|  11.48 |    9.36    | 9.55  |
>
> Performance comparisons (Table below) show Gemma3-4B maintains nearly identical efficiency to VLMLight in handling special vehicles, with only a slight decrease for normal vehicles due to occasional false emergency-vehicle detections (a known hallucination issue).
>
> **Performance Comparison (seconds)**
> |           |      | Songdo | Yau Ma Tei | Massy |
> |:---------:|------|:------:|:----------:|-------|
> | Gemma3-4b | ATT  |  89.81 |    41.38   | 64.79 |
> |           | AETT |  47.54 |    14.57   | 47.01 |
> |  VLMLight | ATT  |  87.14 |    39.80   | 60.84 |
> |           | AETT |  49.88 |    13.50   | 45.41 |
>
> These measures collectively ensure VLMLight meets real-time requirements.
>
> > [Q2]. Could embeddings be used instead of explicit text
>
> We chose text for its explainability and transparency, which is critical for system audits and adoption by traffic authorities. We believe the benefit of human-readable justifications outweighs the potential risk of occasional misleading phrasing-again mitigated by the rule-based action validation step.
>
> > [Q3]. Guaranteeing safety of LLM-issued actions
>
> Safety in VLMLight is ensured through a multi-layer safeguard mechanism:
>
> 1. We have a dual verification of actions. As illustrated in Figure 2, the `Rule Verification Agent` first validates whether the `Plan Agent`'s proposed action complies with traffic regulations. Before execution, a second hard rule check confirms that the action is within the predefined available action set. If it is not, the system automatically falls back to the fast RL branch, ensuring only safe and valid actions are issued.
> 2. Even if an LLM outputs a suboptimal choice, the consequence is limited to minor delays (e.g., longer waiting time), never unsafe outcomes.
> 3. Table 1 demonstrates that LLM agents make correct decisions with high reliability, further supporting their operational safety.
>
> > [Q4]. Framework seems trivial / limited generalization
>
> We clarify that VLMLight is more than a VLM+LLM combination. Its dual-branch architecture merges (i) a high-speed RL policy for standard flow and (ii) a reasoning-based branch for safety-critical cases-mimicking how human traffic officers manage both routine and exceptional situations.
>
> Furthermore, the **open-source simulator is a core contribution that transitions TSC research** from simplified statistics to realistic vision-based control, enabling future generalization to diverse traffic scenarios beyond emergency vehicles.
>
> Reference:
>
> [1] Gu, Yin, et al. "π-light: Programmatic interpretable reinforcement learning for resource-limited traffic signal control." Proceedings of the AAAI Conference on Artificial Intelligence. Vol. 38. No. 19. 2024.
>
> [2] Su, Haoran, et al. "EMVLight: A multi-agent reinforcement learning framework for an emergency vehicle decentralized routing and traffic signal control system." Transportation Research Part C: Emerging Technologies 146 (2023): 103955.
>
> [3] Koonce, Peter. "Traffic signal timing manual. No. FHWA-HOP-08-024." United States. Federal Highway Administration, 2008.
>
> [4] Iryo-Asano, Miho, Wael KM Alhajyaseen, and Hideki Nakamura. "Analysis and modeling of pedestrian crossing behavior during the pedestrian flashing green interval." IEEE Transactions on Intelligent Transportation Systems 16.2 (2014): 958-969.
>
> [5] Government of South Australia, "Traffic Signal Standard: Signal Timings - TS001"
>
> [6] City of Toronto, "Standard Operating Practice: Pedestrian Timing at Signalised Intersections." Transportation Services
>
> [7] Team, Gemma, et al. "Gemma 3 technical report." arXiv preprint arXiv:2503.19786 (2025).

---

> > ### Comment · Reviewer_biB6 · 2025-08-05
> >
> > Appreciate the authors' rebuttal and I believe most of my concerns have been addressed.

---

> > > ### Author Response · Authors · 2025-08-05
> > >
> > > Thank you for your kind response. We truly appreciate your time and constructive feedback throughout the review process.

---

### Official Review · Reviewer_ubqN · 2025-07-02

**Clarity:** 3
**Significance:** 3
**Originality:** 3
**Rating:** 5
**Confidence:** 2

**Summary:**

The paper presents VLMLight, a traffic signal control framework that integrates vision-language meta-control with dual-branch reasoning. At the core of VLMLight is an image-based traffic simulator that enables multi-view visual perception at intersections, allowing policies to reason over rich cues such as vehicle type, motion, and spatial density. A large language model serves as a safety-prioritized meta-controller, selecting between a fast RL policy for routine traffic and a structured reasoning branch for critical cases. Experimental results show that VLMLight reduces the waiting time for emergency vehicles (e.g., ambulances) by up to 65% compared to RL-only baselines, while maintaining comparable performance in standard traffic with less than 1% degradation.

The article contains an introduction, including the traffic signal control, a description of the methodology, setup, and results of experiments, including ablation results. It also contains separate sections presenting the limitations, related works, and broader impacts. The supplementary materials contain the code.

**Questions:**

-

**Ethical Concerns:**

["NO or VERY MINOR ethics concerns only"]

**Final Justification:**

After reading the other reviews and responses, it seems that the issues raised by the other reviewers have been adequately addressed. Therefore, I am maintaining my initial score.

**Limitations:**

There is a separate section describing the identified limitations. I believe they are addresses adequately.

**Paper Formatting Concerns:**

-

**Quality:**

4

**Strengths And Weaknesses:**

Strengths:
- The idea is quite original
- The results of the experiments are very good, the article has the potential to be significant
- The paper is well-written, it has a good structure, and all necessary elements
- There is a good ablation study and discussion of limitations
- The code is available to verify the results
- There is also an analysis of the inference time, suggesting that the method can be applied to real-time traffic signal control

Weaknesses:
- I did not identify any significant weaknesses

---

> ### Author Rebuttal · Authors · 2025-07-31
>
> We sincerely thank the reviewer for their positive evaluation and thoughtful recognition of VLMLight's contributions. As you noted, VLMLight is the first image-based simulator for traffic signal control, enabling multi-view visual perception at intersections and allowing policies to reason over rich visual cues such as vehicle type, motion, and spatial density, an important step beyond traditional RL- or statistics-only approaches.
>
> We also wish to highlight that our vision-language meta-control framework adopts a dual-branch design (fast RL + structured LLM reasoning), enabling the system to manage routine traffic efficiently while reliably handling safety-critical scenarios (e.g., emergency vehicle prioritization) with greater interpretability. This dual-branch reasoning addresses real-world challenges that RL-only methods struggle to manage.
>
> We deeply appreciate your recognition and encouragement, which further reinforces our belief that VLMLight provides a scalable, interpretable, and safety-aware foundation for next-generation traffic signal control.

---

> > ### Comment · Reviewer_ubqN · 2025-08-06
> >
> > Thank you for the rebuttal. I read the other reviews and responses as well. In my opinion, the issues raised by the other reviewers have been adequately addressed, so I'm maintaining my previous score.

---

> > > ### Author Response · Authors · 2025-08-06
> > >
> > > Thank you for taking the time to review our work and read through the rebuttals. We appreciate your recognition and are grateful for your support.

---

### Note · Authors · 2025-08-12

We sincerely thank the reviewers and the AC for their constructive comments and engagement. We have provided detailed responses to all raised concerns during the rebuttal and discussion phase. Below, we summarize the key points and our corresponding clarifications.

- **Textual descriptions vs. embeddings:** In VLMLight, we adopt textual scene descriptions rather than embeddings to ensure interpretability and auditability, both of which are essential for real-world TSC deployment. Moreover, the framework incorporates strict rule-based validation to eliminate any risk from potential noise, ensuring that only feasible actions are passed for implementation.
- **Inference time and scalability:** Referring to government standards for minimum pedestrian green light times, we verified that VLMLight’s reasoning latency stays comfortably within these safety margins. Moreover, multi-agent reasoning is triggered only in rare critical cases, with the fast RL branch handling regular scenarios. Using a more lightweight model such as Gemma3-4B reduces inference time to ~2.5 s even in safety-critical scenarios. Furthermore, the modular design allows seamless integration into multi-intersection coordination without retraining and adapts to varied layouts.
- **Focus on safety-critical events:** Beyond prioritizing emergency vehicles, VLMLight can handle diverse unexpected incidents, such as
objects blocking traffic or workzones. Furthermore, our open-source image-based TSC simulator can model these safety-critical scenarios, enabling the community to develop and evaluate robust vision-based control policies under realistic conditions.

Building on the clarifications provided above, we would like to reiterate the core contributions and practical impact of our work. VLMLight delivers a novel, practical, and safety-aware solution to a high-impact urban mobility challenge. Our dual-branch design, combining vision-language meta-control with a fast RL policy, achieves up to 65% reduction in emergency vehicle waiting times without degrading normal traffic flow performance, while ensuring interpretability, safety guarantees, and real-time deployability. Together with our simulator, which offers the community a realistic platform for advancing TSC research, we believe VLMLight provides both a solid foundation and a forward-looking path for future safe, interpretable, and generalizable traffic control systems.

---

### Decision · Program_Chairs · 2025-09-17

**Decision:**

Accept (poster)

**Comment:**

After the reviews, rebuttal, and discussion two reviewers lean towards reject, two towards acceptance. The reasons for acceptance are:

- A new (open-source) simulator
- Original idea
- A good evaluation of the core ideas

Issues:

- Scalability, performance overheads
- Trade-offs in performance across vehicle types

The authors rebutted both issues well, and the AC sees no major remaining issues that would prevent the paper from being published.